# Spatiotemporal reprogramming of differentiated cells underlies regeneration and neoplasia in the intestinal epithelium

Tsunaki Higa[1], Yasutaka Okita [1], Akinobu Matsumoto [1], Shogo Nakayama[1], Takeru Oka[1], Osamu Sugahara[1], Daisuke Koga [1], Shoichiro Takeishi [1,6], Hirokazu Nakatsumi[1,7], Naoki Hosen[2], Sylvie Robine[3], Makoto M. Taketo[4], Toshiro Sato [5] & Keiichi I. Nakayama [1✉]

Although the mammalian intestinal epithelium manifests robust regenerative capacity after various cytotoxic injuries, the underlying mechanism has remained unclear. Here we identify the cyclin-dependent kinase inhibitor p57 as a specific marker for a quiescent cell population located around the +4 position of intestinal crypts. Lineage tracing reveals that the p57+ cells serve as enteroendocrine/tuft cell precursors under normal conditions but dedifferentiate and act as facultative stem cells to support regeneration after injury. Single-cell transcriptomics analysis shows that the p57+ cells undergo a dynamic reprogramming process after injury that is characterized by fetal-like conversion and metaplasia-like transformation. Population-level analysis also detects such spatiotemporal reprogramming widely in other differentiated cell types. In intestinal adenoma, p57+ cells manifest homeostatic stem cell activity, in the context of constitutively activated spatiotemporal reprogramming. Our results highlight a pronounced plasticity of the intestinal epithelium that supports maintenance of tissue integrity in normal and neoplastic contexts.

[1] Department of Molecular and Cellular Biology, Medical Institute of Bioregulation, Kyushu University, Fukuoka 812-8582, Japan. [2] Department of Hematology and Oncology, Osaka University Graduate School of Medicine, Suita, Osaka 565-0871, Japan. [3] UMR 144, Institut Curie, 75248 Paris Cedex 05, Paris, France. [4] Kyoto University Hospital-iACT (Colon Cancer Project), Graduate School of Medicine, Kyoto University, Kyoto 606-8501, Japan. [5] Department of Organoid Medicine, Keio University School of Medicine, Tokyo 160-8582, Japan. [6] Present address: Department of Cell Biology, Albert Einstein College of Medicine, Bronx, NY 10461, USA. [7] Present address: Department of Molecular Biology, Graduate School of Pharmaceutical Sciences, Nagoya City University, Aichi 467-8603, Japan. ✉email: nakayak1@bioreg.kyushu-u.ac.jp

The rapid turnover and regeneration of the intestinal epithelium in mammals are supported by functionally distinct intestinal stem cell (ISC) populations localized in intestinal crypts. Actively cycling crypt base columnar cells (CBCs) that express Lgr5 at a high-level drive homeostatic epithelial turnover by functioning as a pool of equipotent stem cells while competing for their niche space[1,2]. However, the complete loss of Lgr5[high] CBCs as a result of chemical- or ionizing radiation (IR)-induced injury or of genetic ablation does not prevent maintenance of the intestinal epithelium, suggesting the existence of an alternative ISC population[3,4].

On the basis of the concept that quiescence is a hallmark of undifferentiated tissue stem cells[5], rare and mitotically inactive cells located around the +4 position relative to the intestinal crypt base were proposed to constitute a dedicated and primitive quiescent ISC population[6–10]. However, the existence of such "professional" quiescent ISCs has remained to be demonstrated, given that proposed markers for quiescent +4 ISCs (such as Bmi1, Hopx, Tert, and Lrig1) are also expressed in other crypt cell types including Lgr5[high] CBCs[11,12].

Evidence has suggested that, in response to crypt injury, multiple differentiated cell types including enteroendocrine (EE) cell precursors[13,14], Dll1[+] secretory progenitors[15], Alpi[+] enterocyte progenitors[16], and Paneth cells[17] dedifferentiate and reacquire stem cell potential to act as "facultative" ISCs. Recent studies have also identified two distinct stem cell populations that contribute to postinjury regeneration: Sca1[+] cells[18,19] and Clu[+] revival stem cells (revSCs)[20]. These "intermediary ISCs" are almost undetectable in normal crypts, but emerge specifically after injury and manifest a temporally reverted gene expression profile resembling that of the fetal intestine. Although these cells have been proposed to arise from certain differentiated cell types in the intestinal epithelium, their source and the process by which they are generated have remained largely uncharacterized.

Clinical and histopathologic evidence has revealed that, in the case of almost all epithelial systems of the alimentary tract, a portion of a specific tissue type often spontaneously acquires the histological identity of another tissue type—for example, portions of the intestinal epithelium are frequently replaced with gastric glands and vice versa—a phenomenon referred to as metaplasia[21,22]. Although such metaplasia is generally thought to arise as a result of repetitive tissue injury and the subsequent regeneration process, the physiology of this phenomenon, especially in relation to the tissue stem cell systems, remains poorly understood.

The p57 (CDKN1C) protein is a cyclin-dependent kinase inhibitor and is specifically expressed and required for the maintenance of quiescence and stemness in both hematopoietic[23] and neural[24] stem cells. Here we show that p57 is specifically expressed in quiescent cells located around the +4 position of intestinal crypts. Lineage tracing revealed that these p57[+] cells are EE and tuft cell precursors in normal crypts but that they revert to a stem cell state in response to injury, functioning as facultative ISCs. Single-cell transcriptomics analysis showed that the p57[+] EE/tuft lineage cells acquire a transcriptomic signature reminiscent of that of Clu[+] revSCs and fetal intestine, and they thereby serve as a source of intermediary ISCs that arise during postinjury regeneration. Concomitantly with such fetal-like reversion, we found that the p57[+] cells undergo a metaplasia-like transformation during the postinjury reprogramming process that is characterized by the acquisition of a transcriptomic signature similar to that of adult gastric epithelium. Further population- and histological level analyses revealed that such "spatiotemporal reprogramming" also occurs widely in cells other than p57[+] cells, both in regenerating and in adenomatous crypts. Our results thus highlight the pronounced plasticity of the intestinal epithelium that supports the maintenance of tissue integrity, and they provide insight into the pathological relevance of such plasticity in normal and neoplastic contexts.

## Results

### Expression and function of p57 in the intestinal epithelium.
We previously showed that p57 is expressed in hematopoietic and neural stem cells with marked specificity, and is essential for the maintenance of the quiescence and stemness of these cells[23,24]. We therefore postulated that p57 might also serve as a marker for the specific identification of reserve or quiescent ISC populations. To investigate the expression pattern and functional role of p57 in intestinal crypts, we performed immunofluorescence staining of the mouse small intestine. We found that p57 expression is restricted to rare cells positioned immediately above the Lgr5[high] CBC and Paneth cell zone (Fig. 1a), corresponding to the +4 position relative to the crypt base (Fig. 1b). These p57[+] cells were all negative for the proliferation marker Ki67 (Fig. 1c, d), suggesting that p57 marks quiescent cells around the +4 position of intestinal crypts. The p57[+] cells overlapped with the vast majority of cells expressing Bmi1 (Fig. 1e, f) and with a subset of crypt label-retaining cells (LRCs) (Fig. 1g), both of which have been proposed to represent reserve or facultative ISC populations[3,13]. Given that the numerous marker genes proposed for quiescent ISC populations have been shown to be expressed also in Lgr5[high] CBCs[12], we next examined the specificity of p57 expression for Bmi1[high] cells at the transcript level. In fluorescence-activated cell sorting (FACS) analysis, the Bmi1-GFP[high] fraction (~1%) isolated from heterozygous Bmi1–GFP (green fluorescent protein) knock-in mice[3] and the Lgr5-GFP[high] fraction (~4%) isolated from heterozygous Lgr5-EGFP-IRES-CreER[T2] knock-in mice[1] were regarded as Bmi1[high] cells and Lgr5[high] CBCs, respectively (Fig. 1h). Reverse transcription (RT) and quantitative polymerase chain reaction (qPCR) analysis revealed that the p57 gene was expressed almost exclusively in Bmi1[high] cells, being virtually silent in Lgr5[high] CBCs, whereas the classical +4 markers (Bmi1, Lrig1, Tert, and Hopx) were substantially expressed also in Lgr5[high] CBCs (Fig. 1i).

To investigate the functional role of p57 in the maintenance of quiescence in these p57[+] cells, we next induced its genetic ablation in mice. The p57 gene locus undergoes genomic imprinting, with only the maternal allele being expressed. To disrupt the maternal allele of the p57 gene only in the intestinal epithelium, we crossed female mice harboring a floxed allele of the p57 gene (p57[+/F])[23] with male mice harboring both a tamoxifen-inducible CreER[T2] transgene driven by the Villin gene promoter (Villin-CreER[T2])[25] and the Bmi1-GFP knock-in allele (p57 CKO mice) (Fig. 2a, b). We injected control and p57 CKO mice with 5-ethynyl-2′-deoxyuridine (EdU) to examine the frequency of Bmi1[high] cells in S phase. EdU incorporation after 6 h was almost undetectable in Bmi1[high] crypt cells of control mice, whereas the frequency of EdU[+]Bmi1[high] cells was markedly increased in p57 CKO mice (Fig. 2c, d). We also performed flow cytometric analysis of the Bmi1[high] fraction of intestinal epithelial cells stained for Ki67 and with propidium iodide (PI), and found that the proportion of p57-deficient cells in $G_0$ phase of the cell cycle was markedly reduced compared with that for control cells (Fig. 2e, f). Together, these results suggested that p57 specifically identifies and contributes to the maintenance of the quiescence of a mitotically inactive cell type around the +4 position of intestinal crypts, and that it might therefore be a promising candidate as a marker for reserve or facultative ISCs.

### p57 marks quiescent EE/tuft lineage cells.
To characterize further the relations between p57[+] cells and other intestinal cell

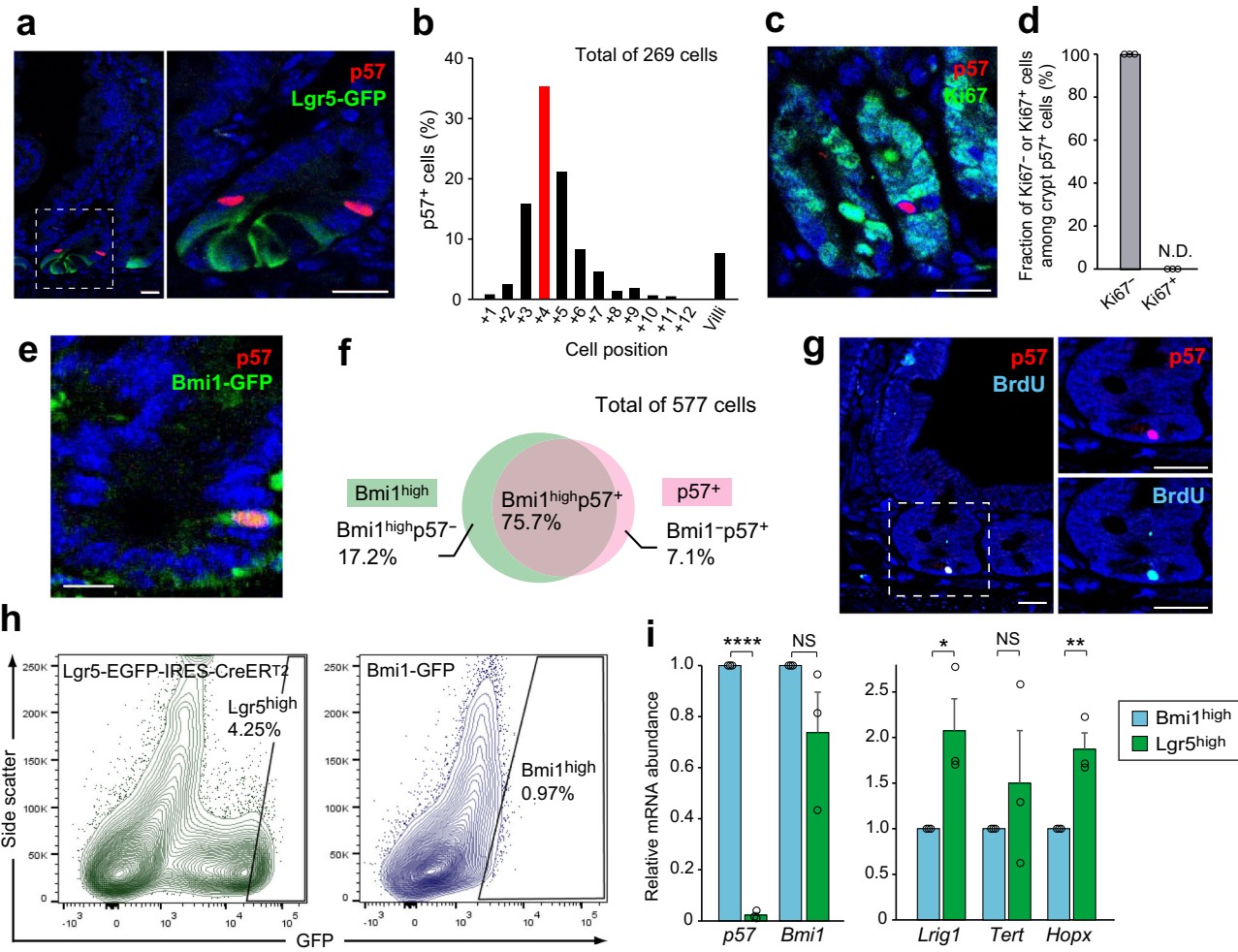

**Fig. 1 p57 is specifically expressed in quiescent cells around the +4 position of intestinal crypts. a** Representative immunofluorescence staining of p57 and Lgr5-GFP in the intestine of an Lgr5-EGFP-IRES-CreER[T2] knock-in mouse. The boxed region in the left panel is shown at higher magnification in the right panel. Arrowheads indicate p57+ cells. **b** Quantification of the number of p57+ cells at each position of crypts. **c** Representative immunofluorescence staining of p57 and Ki67 in the intestine of a wild-type (WT) mouse. The arrowhead indicates a p57+ cell. **d** Quantification of the fraction of Ki67+ or Ki67− cells among p57+ cells. A total of 157 p57+ crypt cells was counted from three mice. N.D., not detected. **e** Representative immunofluorescence staining of p57 and Bmi1-GFP in the intestine of a Bmi1-GFP knock-in mouse. The arrowhead indicates a p57+Bmi1[high] cell. **f** Venn diagram showing the frequency of Bmi1[high]p57−, Bmi1−p57+, and Bmi1[high]p57+ cells in the duodenum and jejunum. **g** Representative immunofluorescence staining of p57 and bromodeoxyuridine (BrdU) in the intestine of a WT mouse subjected to a 10-day BrdU label-retention assay. The boxed region containing a p57+BrdU+ cell (arrowhead) in the left panel is shown at higher magnification in the right panels. **h** FACS gating strategy for sorting and analysis of Lgr5[high] CBCs (left) and Bmi1[high] cells (right) from the intestinal epithelium. **i** RT-qPCR analysis of relative mRNA abundance for p57 and putative +4 position markers in Bmi1[high] cells and Lgr5[high] CBCs isolated as in **h** (n = 3 mice). Quantitative data in **d** and **i** are means + SEM. NS (not significant), *P < 0.05, **P < 0.01, ****P < 0.001 (two-tailed Student's t test). Scale bars: 20 μm (**a**, **c**, **g**) or 10 μm (**l**). Source data are provided as a Source Data file.

types, we performed comparative analysis with a single-cell RNA-sequencing (scRNA-seq) data set[14] for Bmi1[high] or Prox1+ cells, Lgr5[high] CBCs, and Lgr5− cells (Fig. 3a). Principal component analysis and subsequent t-distributed stochastic neighbor embedding (t-SNE) with the use of the single-cell analysis package Seurat[26] divided these single cells into 15 unsupervised clusters (Supplementary Fig. 1a, b) characterized by enrichment of known cell type-specific markers (Supplementary Fig. 1a). Lgr5[high] CBCs were divided into discrete subpopulations on the basis of their Ki67 expression levels (Supplementary Fig. 1c), as well as on cell cycle scoring analysis (Supplementary Fig. 1d), with these subpopulations being designated cycling and non-cycling CBCs (Fig. 3a).

Consistent with our population-level analysis (Fig. 1i), we found that the p57 gene was expressed exclusively in Bmi1[high] or Prox1+ populations (Fig. 3b), with such expression being almost completely absent in clusters containing Lgr5[high] CBCs or the other lineage-committed progenitor cell types. Given our initial notion that p57 might identify a reserve or quiescent ISC population, we examined the expression of quiescent stem cell signature genes (Supplementary Table 1), which show common expression kinetics in the transcriptomes of three representative types of quiescent tissue stem cells: hematopoietic, muscle, and hair follicle stem cells[5]. We found that the expression patterns of both up-regulated and down-regulated signature genes in these quiescent stem cells were more similar to those in the Bmi1[high] or Prox1+ populations than to those in cycling or noncycling CBCs (Fig. 3c), indicative of the quiescent state of these former populations. Nevertheless, p57-expressing Bmi1[high] or Prox1+ populations showed marked expression of EE/tuft cell marker genes (Fig. 3d and Supplementary Fig. 1e) as well as of signature genes or individual markers (Peg3, Pax6, and Rfx6) of intestinal

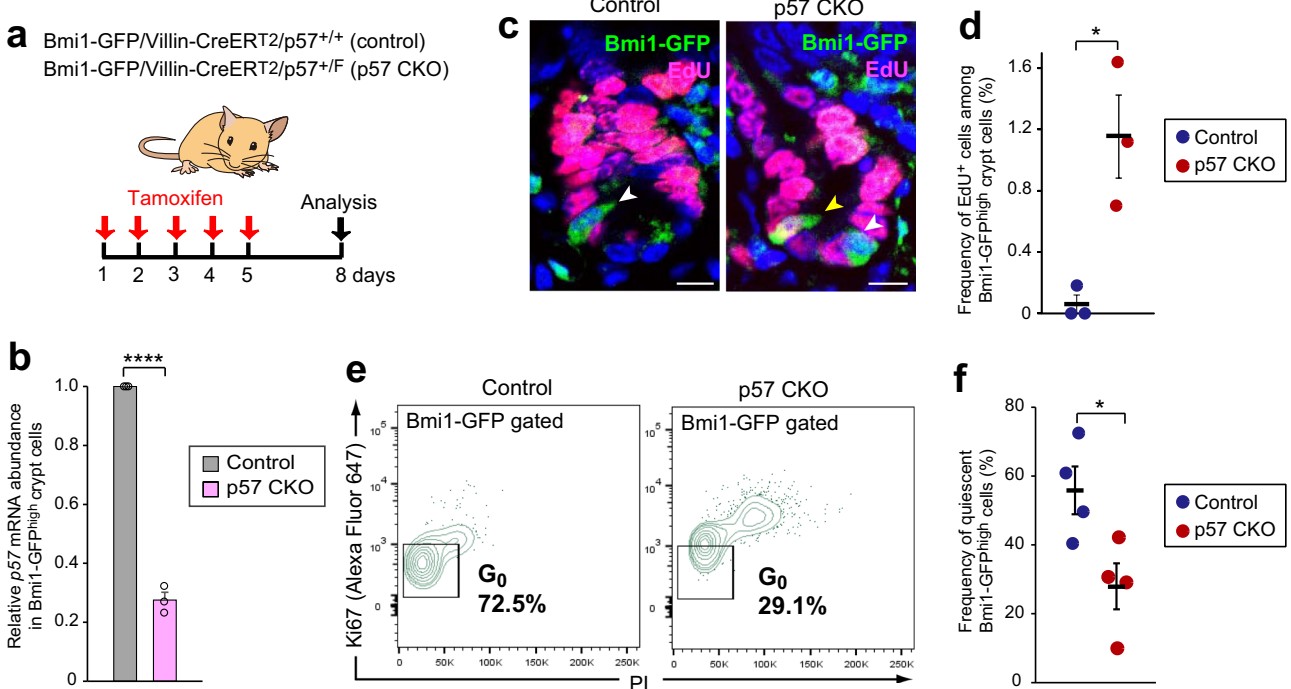

**Fig. 2 p57 is required for the maintenance of quiescence in Bmi1$^{high}$ crypt cells. a** Experimental setup for deletion of the maternal allele of the p57 gene in the small intestine. Bmi1-GFP/Villin-CreER$^{T2}$/p57$^{+/+}$ (control) and Bmi1-GFP/Villin-CreER$^{T2}$/p57$^{+/F}$ (p57 CKO) mice were injected intraperitoneally with tamoxifen (50 mg/kg) on 5 consecutive days and were analyzed 3 days after the final injection. **b** RT-qPCR analysis of relative p57 mRNA abundance in Bmi1$^{high}$ crypt cells isolated as in Fig. 1h from the small intestine of control or p57 CKO mice ($n = 3$ mice). **c** Representative immunofluorescence staining of EdU and Bmi1-GFP in the intestine of control and p57 CKO mice 6 h after intraperitoneal injection of EdU (20 mg/kg). Yellow and white arrowheads indicate Bmi1-GFP$^{high}$ cells positive or negative for EdU, respectively. Scale bar, 10 μm. **d** Quantitation of the frequency of EdU$^+$ cells among Bmi1-GFP$^{high}$ crypt cells as in **c** ($n = 3$ mice). **e** Representative flow cytometric analysis of the frequency of quiescent (Ki67$^-$PI$^-$) cells in the Bmi1-GFP$^{high}$ fraction of intestinal epithelial cells from control and p57 CKO mice treated as in **a**. **f** Quantitative analysis for determinations as in **e** ($n = 4$ mice). Quantitative data in **b** are means + SEM, and those in **d** and **f** are means ± SEM. *$P < 0.05$, **$P < 0.01$, ****$P < 0.001$ (two-tailed Student's $t$ test). Source data are provided as a Source Data file.

LRCs (Fig. 3e and Supplementary Fig. 1f), which have been shown to represent EE cell progenitors in the steady state but to acquire stem cell activity in response to injury[13].

To visualize and characterize the crypt p57$^+$ cells experimentally, we established mice harboring an in-frame knock-in of the diphtheria toxin receptor (DTR) and Venus cDNA (each preceded by cDNA for the self-cleaving T2A peptide) linked to the 3' portion of the endogenous p57 coding sequence (Fig. 3f). Immunofluorescence analysis revealed that Venus expression in mice harboring the maternally inherited p57-DTR-Venus allele recapitulated precisely the expression pattern of the endogenous p57 protein in intestinal crypts (Fig. 3g). FACS and subsequent RT-qPCR analysis confirmed that p57 mRNA was exclusively enriched in the upper 0.5% to 0.7% of the Venus$^+$ fraction among total crypt cells (Fig. 3h and Supplementary Fig. 2a), suggesting that this reporter mouse line allows specific labeling of crypt p57$^+$ cells. RT-qPCR analysis of the sorted p57-Venus$^+$ cells and total crypt cells also revealed that the p57$^+$ cells preferentially expressed representative LRC-specific genes and EE/tuft cell marker genes (Fig. 3h), consistent with the findings of the scRNA-seq analysis (Fig. 3d, e). Expression of classical +4 ISC marker genes such as *Bmi1*, *Lrig1*, and *Hopx* was not significantly enriched in the p57-Venus$^+$ cells (Supplementary Fig. 2b), likely as a result of the substantial expression of these genes in other crypt cell types including Lgr5$^{high}$ CBCs[12] (Fig. 1i). Collectively, these results suggested that, although p57-expressing cell populations share the quiescence gene program with other tissue stem cells, they can be classified as EE/tuft lineage cells rather than as undifferentiated, "professional" quiescent ISCs.

**p57$^+$ cells serve as injury-induced facultative ISCs**. Given that p57 was found to specifically mark quiescent EE/tuft lineage cells located around the crypt +4 position, we examined the stem cell potential of these cells by lineage tracing based on the p57 gene promoter in vivo. We generated mice that harbor an in-frame T2A-CreER$^{T2}$ cassette knocked-in immediately 3' of the p57 coding sequence (p57-CreER$^{T2}$ mice). Female p57-CreER$^{T2}$ mice were then crossed with male CAG–loxP-Stop-loxP (LSL)–tdTomato reporter mice[27] (Fig. 4a). At 24 h after a single tamoxifen injection (50 mg/kg) in p57-CreER$^{T2}$/CAG-LSL-tdTomato mice, tdTomato$^+$ cells emerged around the +4 position of a subset of small intestinal crypts (Fig. 4b, c) at a frequency of ~1 cell per 100 crypts (Supplementary Fig. 2c). Comparison with the frequency of p57-Venus$^+$ cells or endogenous p57 protein-expressing cells (~1 cell per 20 crypts) suggested that 20% to 25% of all p57$^+$ cells in intestinal crypts were labeled in this lineage tracing system (Supplementary Fig. 2c). Consistent with our results in Fig. 1, these tdTomato$^+$ cells at 24 h after tamoxifen injection were completely negative for labeling with EdU in short-term pulse-chase analysis (Fig. 4d, e). At 7 days after tamoxifen injection, solitary tdTomato$^+$ cells expressing Chga (Fig. 4f) or Dclk1 (Fig. 4g) were frequently observed in villi, suggesting that the p57$^+$ cells in crypts serve as EE/tuft cell progenitors, consistent with their gene expression patterns in Fig. 2. However, the formation of a "tracing ribbon" emanating from the crypt base to the top of the villus was essentially undetectable in both short-term (1–2 weeks) and long-term (1 to 12 months) tracing experiments (Fig. 4j), suggesting that the p57$^+$ cells do not contribute to the homeostatic turnover of the intestinal epithelium or maintenance of the ISC pool.

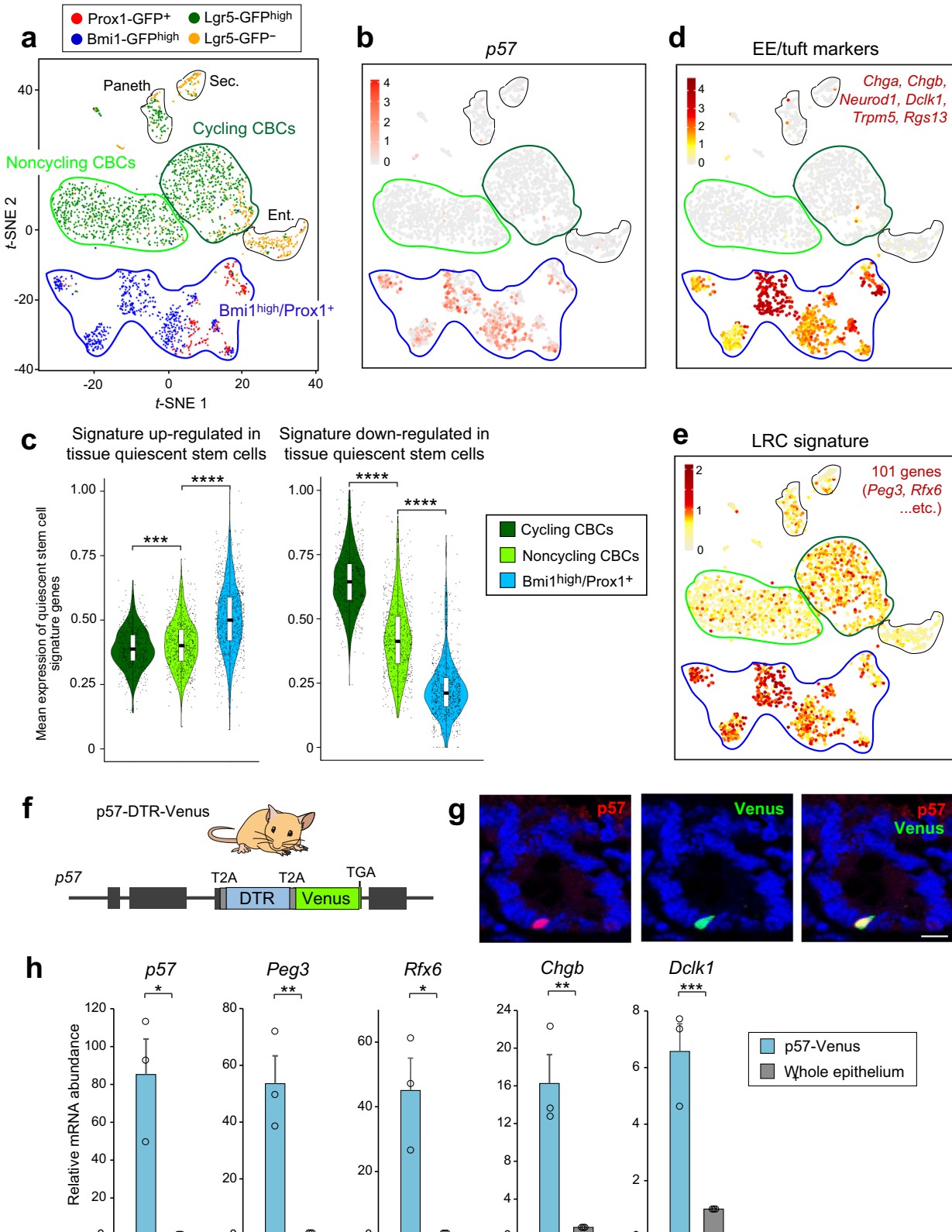

We next examined whether the p57+ cells serve as a reserve ISC population for postinjury epithelial regeneration. We injected p57-CreER$^{T2}$/CAG-LSL-tdTomato mice with tamoxifen as well as either injected them with 5-fluorouracil (5-FU, 150 mg/kg) or exposed them to IR (6 Gy), both of which induce injury associated with the almost complete elimination of CBCs[4] (Supplementary Fig. 2d), with analysis being performed at 14 days after treatment (Fig. 4h). As an alternative injury context, p57-CreER$^{T2}$/CAG-LSL-tdTomato mice also harboring an Lgr5-DTR-EGFP cassette[3] were injected with tamoxifen, as well as with diphtheria toxin (DT) in order to ablate the Lgr5-expressing cell population containing CBCs (Fig. 4h). In marked contrast to the results obtained for the steady state, lineage tracing events were detected in all regions of the small intestine of all mice subjected

**Fig. 3 p57 marks quiescent EE/tuft lineage cells. a** t-SNE projection of 1416 Lgr5-GFP[high] (green), 611 Bmi1-GFP[high] (blue), 206 Prox1-GFP[+] (red), and 407 Lgr5-GFP[−] (orange) crypt cells of the mouse intestine based on scRNA-seq data[14]. Cycling CBCs, noncycling CBCs, and Bmi1[high]/Prox1[+] cell groups are enclosed in color-coded frames. Enterocyte (Ent.), secretory cell (Sec.), and Paneth cell groups are enclosed in black frames. **b** Heat map for the abundance of p57 mRNA on the t-SNE projection. **c** Violin plots for the relative mean expression levels in cycling CBCs, noncycling CBCs, and Bmi1[high]/Prox1[+] cells of signature genes up-regulated or down-regulated in quiescent stem cells[5]. The overlaid box plots represent the maximum, upper quantile, lower quantile and minimum values. Bars correspond to the median values. ***$P < 0.005$, ****$P < 0.001$ (Steel-Dwass test). **d, e** Heat maps for the mean expression levels of the indicated EE/tuft cell marker genes (**d**) and of LRC signature genes[13] (**e**) shown on the t-SNE projection. **f** Schematic representation of the p57 gene locus of p57-DTR-Venus mice. **g** Representative immunofluorescence staining of p57 and Venus protein in the intestine of p57-DTR-Venus mice. Scale bar, 10 μm. **h** RT-qPCR analysis of relative mRNA abundance for LRC or EE/tuft cell markers in sorted p57-Venus[+] cells and total crypt epithelial cells. Data are means + SEM ($n = 3$ mice). *$P < 0.05$, **$P < 0.01$, ***$P < 0.005$ (two-tailed Student's t test). Individual genes for each signature are listed in Supplementary Table 1. Source data are provided as a Source Data file.

to 5-FU- or IR-induced injury or to genetic ablation of CBCs (Fig. 4i, j, and Supplementary Fig. 2e). The frequencies of these events in the duodenum, jejunum, and ileum were similar. The 5-FU–induced tracing ribbons contained all three major differentiated cell types (enterocytes, goblet cells, and Paneth cells) (Supplementary Fig. 2f–h) and, importantly, Lgr5[high] CBCs (Fig. 4k). These results suggested that the quiescent p57[+] cells in crypts serve as EE/tuft cell precursors in the steady state, but that they facultatively revert to functional ISCs and give rise to heterogeneous intestinal cell types in response to crypt damage.

We observed that 5-FU treatment resulted in a significant activation of Wnt signaling in intestinal crypts (Supplementary Fig. 3a), which plays a key role not only in the maintenance but also in the dedifferentiation-mediated induction of ISCs[15,16]. To examine directly the responsiveness of p57[+] cells to Wnt signaling, we assessed the formation of intestinal organoids[28] in vitro from p57[+] cells isolated from p57-CreER[T2]/CAG-LSL-tdTomato mice at 24 h after tamoxifen treatment (Supplementary Fig. 3b). We found that p57[+] cells completely lacked the ability to form organoids under the standard organoid culture condition with ENR medium (containing epidermal growth factor, Noggin, and R-spondin-1). In contrast, we detected the formation of fully tdTomato-labeled organoids in the presence of either Wnt3a (ENRW medium) or the Wnt signaling activator CHIR99021 (ENRC medium). This activity was even more pronounced in the presence of both Wnt3a and CHIR99021 (ENRWC medium) (Supplementary Fig. 3c, d). In addition, we detected p57[+] cell-derived lineage tracing events in the small intestine of p57-CreER[T2]/CAG-LSL-tdTomato mice injected with CHIR99021 (Fig. 4j), which has been shown to activate Wnt signaling in the intestinal epithelium in vivo[29]. The lower frequency of CHIR99021-induced lineage tracing events compared with those associated with injury contexts might be attributable to the lack of long-term ISC niche availability, given that CHIR99021 is likely to elicit a transient activation of Wnt signaling in p57[+] cells but does not eliminate the preexisting Lgr5[high] CBCs. These results suggested that postinjury activation of Wnt signaling contributes, at least in part, to the reversion of p57[+] cells to functional ISCs.

To examine whether p57[+] cells are required for postinjury intestinal regeneration, we induced ablation of these cells by injecting DT (50 μg/kg) into p57-DTR-Venus mice. Unexpectedly, however, we found that almost all p57-DTR-Venus mice died within 48 h after DT treatment (Supplementary Fig. 3e) without any apparent intestinal phenotype, thus precluding long-term assessment of these mice. We therefore investigated the effect of p57[+] cell ablation in vitro with the use of organoids established from the small intestine of p57-DTR-Venus mice. After initial culture in ENR medium with or without 5-FU at 1 μg/ml (to induce injury) or DT at 5 ng/ml (to induce p57[+] cell ablation) according to the indicated time schedule (Fig. 4l and Supplementary Fig. 3f), the p57-DTR-Venus organoids were dissociated into single cells and replated for quantification of

organoid formation efficiency. Pretreatment with 5-FU alone resulted in the formation of spherelike undifferentiated intestinal organoids[18] (Fig. 4m) with an efficiency that was not significantly different from that for the control condition (Fig. 4n), suggesting that the organoids were regenerated by differentiated cells that had undergone postinjury reversion, even though CBCs were lost to injury. Organoid formation efficiency was markedly reduced, however, after pretreatment with the combination of 5-FU and DT (Fig. 4n) compared with that with 5-FU alone, indicating that ablation of p57[+] cells impaired postinjury regeneration of organoids. Together with our lineage tracing experiments, these findings suggested that p57[+] cells substantially contribute to postinjury regeneration of the intestinal epithelium through injury-induced dedifferentiation.

**Spatiotemporal reprogramming in regenerating p57[+] cells.** Our tracing experiments indicated that quiescent p57[+] cells dedifferentiate from their EE/tuft cell precursor state and replenish Lgr5[high] CBCs lost as a result of crypt injury. We next investigated population-wide transcriptomic changes underlying this reversion process. We thus performed the scRNA-seq analysis with the CEL-seq2 platform[30] of p57-Venus[+] and p57-Venus[−] cells isolated from crypts of the duodenum, jejunum, and ileum of p57-DTR-Venus mice, both under the normal condition (3 days after vehicle treatment) and during postinjury regeneration (3 days after 5-FU treatment) (Fig. 5a and Supplementary Fig. 4a). For each condition, two biological replicates were obtained from distinct mouse samples (replicates 1 and 2). Clustering of the cells was performed with the use of Seurat in combination with Harmony software[31] (Supplementary Fig. 4b), an integration tool for scRNA-seq data sets, which allowed us to avoid batch effect-driven clustering apparent in simple t-SNE analysis (Supplementary Fig. 4c). The normal and regenerating crypt cells were divided into 20 distinct clusters on t-SNE projection (Fig. 5b), with the clusters being characterized by the expression of known cell-type-specific genes (Supplementary Fig. 4d, e). The postinjury clusters showed relative enrichment of tuft and Paneth cells compared with the normal clusters (Fig. 5b), with the latter finding being consistent with a previous scRNA-seq study of the mouse intestinal epithelium after exposure to IR[16]. Of note, cluster 3 consisted of cells expressing both Paneth cell and secretory cell (that is, secretory precursor or goblet lineage cell) markers, indicating that these cells are in an intermediary state between these cell types.

The normal p57-Venus[+] cells consisted mostly of EE/tuft cells with a few Paneth cells (Fig. 5b and Supplementary Fig. 4e). In addition, our earlier findings showed that p57[+] cells overlap with LRCs (Figs. 1g and 3e). These results are consistent with a previous study showing that intestinal LRCs are precursors of both EE and Paneth cells[13]. However, given that we did not observe significant enrichment of Paneth markers in p57[+] cells by RT-qPCR analysis (Supplementary Fig. 2b), we considered

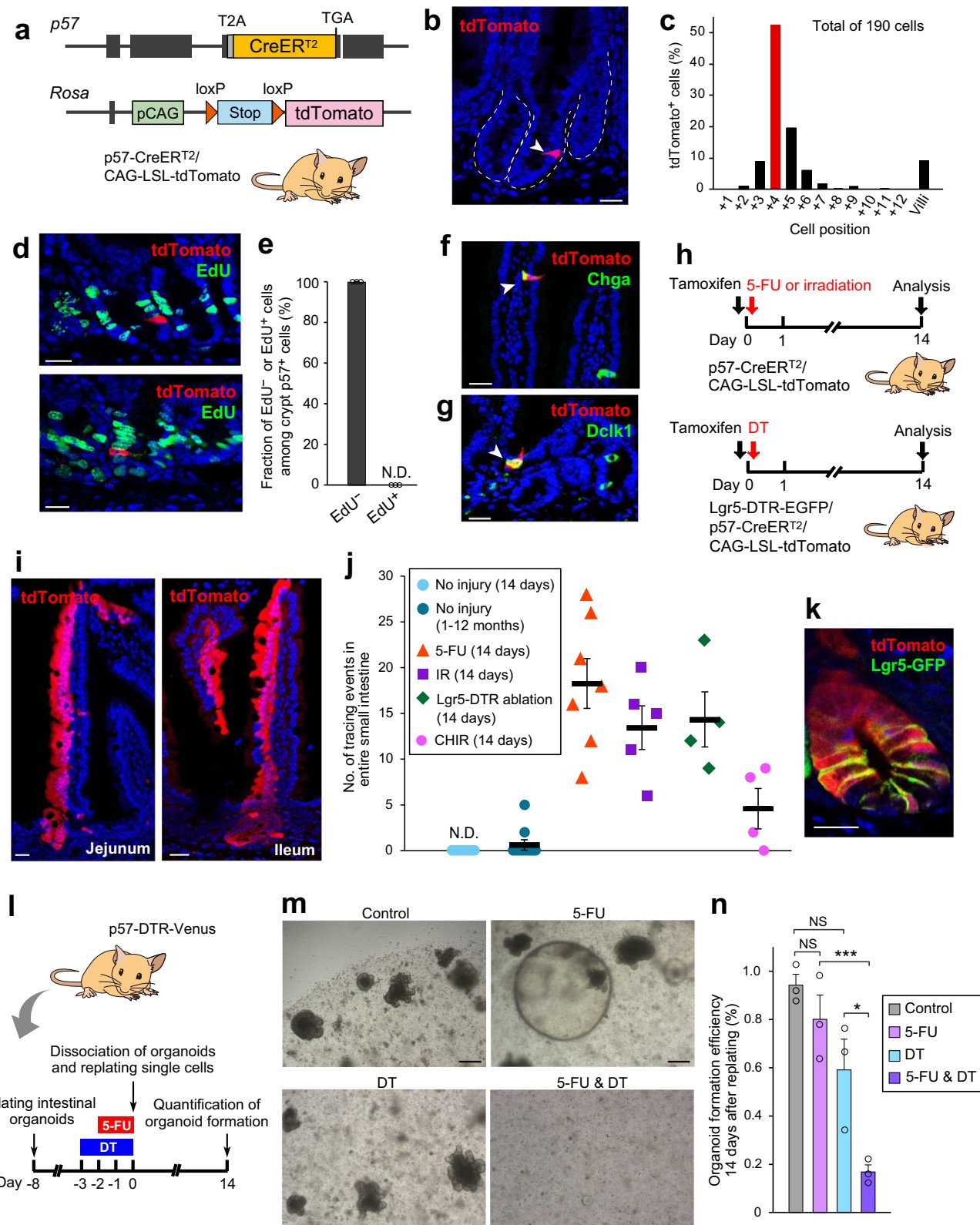

p57+ cells as EE/tuft precursors. Of note, p57 mRNA was detected specifically in normal p57-Venus+ clusters, being seldomly detected in postinjury p57-Venus+ clusters (Supplementary Fig. 4f) in this data set. This finding is presumably attributable to the partial down-regulation of p57 gene expression in association with transient mitotic activation (Supplementary Fig. 4g) or with alteration of cell identity after injury, which might

have resulted in a decline in transcript abundance to below the threshold for detection by scRNA-seq analysis. On the other hand, the abundance of the Venus protein was maintained even in postinjury p57+ cells (Supplementary Fig. 4a), which is likely due to its marked stability in mammalian cells[32]. Consistent with this notion, RT-qPCR analysis of the cells subjected to our scRNA-seq analysis showed that p57 gene expression was

**Fig. 4 Quiescent p57+ cells serve as facultative ISCs. a** Schematic representation of the p57 and Rosa gene loci of p57-CreER[T2]/CAG-LSL-tdTomato mice. **b** Representative immunofluorescence staining of tdTomato-labeled p57+ cells in the intestine. **c** Quantification of the number of tdTomato-labeled p57+ cells at each crypt position. **d** Representative immunofluorescence images of tdTomato-labeled p57+ cells and EdU in the intestine of mice subjected to 2-h EdU pulse-chase analysis. **e** Quantification of the fraction of EdU+ or EdU− cells among 316 tdTomato-labeled p57+ crypt cells from 3 mice. Data are means + SEM. N.D., not detected. **f, g** Representative immunofluorescence images of tdTomato-labeled intestinal cells (arrowheads) positive for Chga (**f**) or Dclk1 (**g**). **h** Dosing regimen for mice subjected to lineage tracing of p57+ cells in combination with 5-FU or radiation exposure (top) or Lgr5[high] cell ablation (bottom). **i** Representative immunofluorescence images of tracing events for p57+ cells in the jejunum (left) and ileum (right) of mice subjected to 5-FU-induced injury. **j** Quantification of tracing events for p57+ cells under the indicated experimental conditions. Tracing was performed for 14 days ($n = 7$ mice) or for 1 to 12 months ($n = 10$) under the homeostatic condition and for 14 days in mice either exposed to 5-FU at 150 mg/kg ($n = 7$), 6 Gy of IR ($n = 5$), or CHIR99021 at 1 mg/kg ($n = 4$) or subjected to ablation of Lgr5[high] cells ($n = 4$). Data are means ± SEM. N.D., not detected. **k** Representative immunofluorescence image of 5-FU-induced p57+ cell-derived clones containing Lgr5-GFP[high] CBCs. **l** Experimental protocol for evaluation of the effect of p57+ cell ablation in intestinal organoids (see Methods for details). **m** Representative phase-contrast images of p57-DTR-Venus organoids at 14 days after replating as in **l**. **n** Efficiency of organoid formation quantified from images as in **m**. Data are means + SEM ($n = 3$ independent experiments). NS (not significant), *$P < 0.05$, ***$P < 0.005$ (two-tailed Student's $t$ test). Scale bars, 20 μm (**b, d, f, g, i, k**) or 100 μm (**m**). Source data are provided as a Source Data file.

partially reduced but remained predominant in the Venus+ fraction after injury (Supplementary Fig. 4h), suggesting that the scRNA-seq data set based on p57-DTR-Venus mice faithfully recapitulates the kinetics of p57+ cells undergoing regeneration.

Intriguingly, most of the postinjury cell populations were not characterized decisively by the expression of typical intestinal cell type-specific markers (Supplementary Fig. 4e). We therefore classified these cells into two groups: Regenerating 1 (clusters 4, 10, 17, and 18), which consisted mostly of postinjury p57-Venus+ cells; and Regenerating 2 (clusters 1, 2, and 15), which consisted mainly of postinjury p57-Venus− cells. Regenerating 1 cells manifested marked enrichment of the signature genes of Clu+ revSCs (Fig. 5c), which were recently shown to emerge specifically after injury and to act as intermediary ISCs fueling subsequent regeneration[20]. This result suggested that the p57+ EE/tuft lineage cells acquire the revSC profile in response to injury and constitute at least one of the preexisting sources for such intermediary ISCs. To characterize further the transcriptomic changes, we performed an unbiased analysis of differentially expressed genes for the clusters constituting Regenerating 1, as well as the other clusters. Unexpectedly, we found that the top-hit differentially expressed genes up-regulated in Regenerating 1 cells were enriched not only with markers (including *Clu*, *Mif*, and *Ly6a*) for revSCs[20] or fetal intestine[18,19] but also with a series of genes specific for adult gastric cell types (Fig. 5d). The latter genes included those for gastrokines (*Gkn2* and *Gkn3*), secretory peptides derived from foveolar cells; that for pepsinogen C (*Pgc*), a proteolytic zymogen secreted from chief cells; those for stomach-type trefoil factors and mucin (*Tff1/2* and *Muc6*, respectively) secreted from gastric mucous cells; those for stomach-type cathepsin and claudin (*Ctse* and *Cldn18*, respectively); and that for aquaporin 5 (*Aqp5*), a recently identified marker for gastric stem cells in pyloric glands. In contrast, the top-hit differentially expressed genes downregulated in Regenerating 1 cells included multiple genes specific for adult intestine (Fig. 5d), some of which are characteristic of EE/tuft or secretory cells (*Cdhr2/5* and *Tff3*). On the *t*-SNE projection, a previously published gene set characteristic of fetal intestine[33] (Supplementary Table 1) was specifically expressed in Regenerating 1 cells (Fig. 5e and Supplementary Fig. 5a, b), suggestive of temporal reversion to the fetal-like state. In addition, the expression of a gene signature for adult gastric epithelium[34] (Supplementary Table 1) was also concomitantly upregulated in this population exclusively and to a marked extent (Fig. 5f and Supplementary Fig. 5a, c). Consistent with the upregulation of gastric signature genes, we found that *Nkx6-3*, which encodes a transcription factor that plays an essential and deterministic role in gastric epithelial differentiation[35,36], was expressed in Regenerating 1 cells (Fig. 5g).

This population also showed significant enrichment of *Ascl2* expression (Fig. 5h), which is characteristic of and confers stemness on Lgr5[high] CBCs in normal crypts[37], suggestive of the acquisition of an ISC-like state concomitantly with the transcriptomic changes described above. The observed enrichment of gastric gene expression is indicative of the transformation of spatial histological identity and reminiscent of gastric metaplasia, which is often apparent clinically in association with intestinal inflammatory conditions such as duodenal ulcer and Crohn's disease[22,38]. The up-regulation of representative marker genes for fetal intestine and adult stomach in postinjury p57-Venus+ cells was also confirmed at the population level by RT-qPCR analysis (Fig. 5i). Together, these results indicated that, during their injury-induced transition to the ISC state, p57+ EE/tuft lineage cells undergo both fetal-like reversion and metaplasia-like transformation of their transcriptomic profile, a phenomenon we refer to as spatiotemporal reprogramming.

**Acquisition of stem cell-like transcriptome in p57+ cells.** To evaluate the degree of "stemness" of each cell population in our scRNA-seq data set, we performed an analysis with CytoTRACE[39] (Fig. 6a and Supplementary Fig. 5d), an algorithm that infers stem cell properties by scoring each cell on the basis of its transcriptional diversity. Across all cell clusters, the top three clusters—clusters 12, 8, and 6—corresponded to CBC and some enterocyte populations of normal p57-Venus− cells (Fig. 6a and Supplementary Fig. 5e), suggestive of a high degree of stemness for these cell types in the normal intestinal crypt. The normal p57+ cells (including clusters 0, 9, and 13, for example) showed low levels of stemness, consistent with the lack of stem cell activity apparent in the lineage tracing analysis at steady state. However, among postinjury cell populations, clusters 4 and 15, which belong to the p57+ cell-enriched Regenerating 1 group, showed the highest CytoTRACE scores (Fig. 6a and Supplementary Fig. 5e). Comparison of the scores according to p57-Venus expression status and condition revealed that postinjury p57-Venus+ cells showed significantly higher scores than did normal p57-Venus+ or postinjury p57-Venus− cells (Fig. 6b). These results suggested that p57+ cells serve as a predominant ISC population during regeneration as a result of their injury-induced acquisition of stemness.

Consistent with these findings, we found that p57-Venus+ crypt cells sorted from the small intestine of mice at 3 days after 5-FU injection were able to form intestinal organoids (Fig. 6c, d), whereas those from noninjured mice almost completely lacked this ability. Together with our findings presented in Fig. 4, these results suggested that p57+ crypt cells acquire stem cell-like

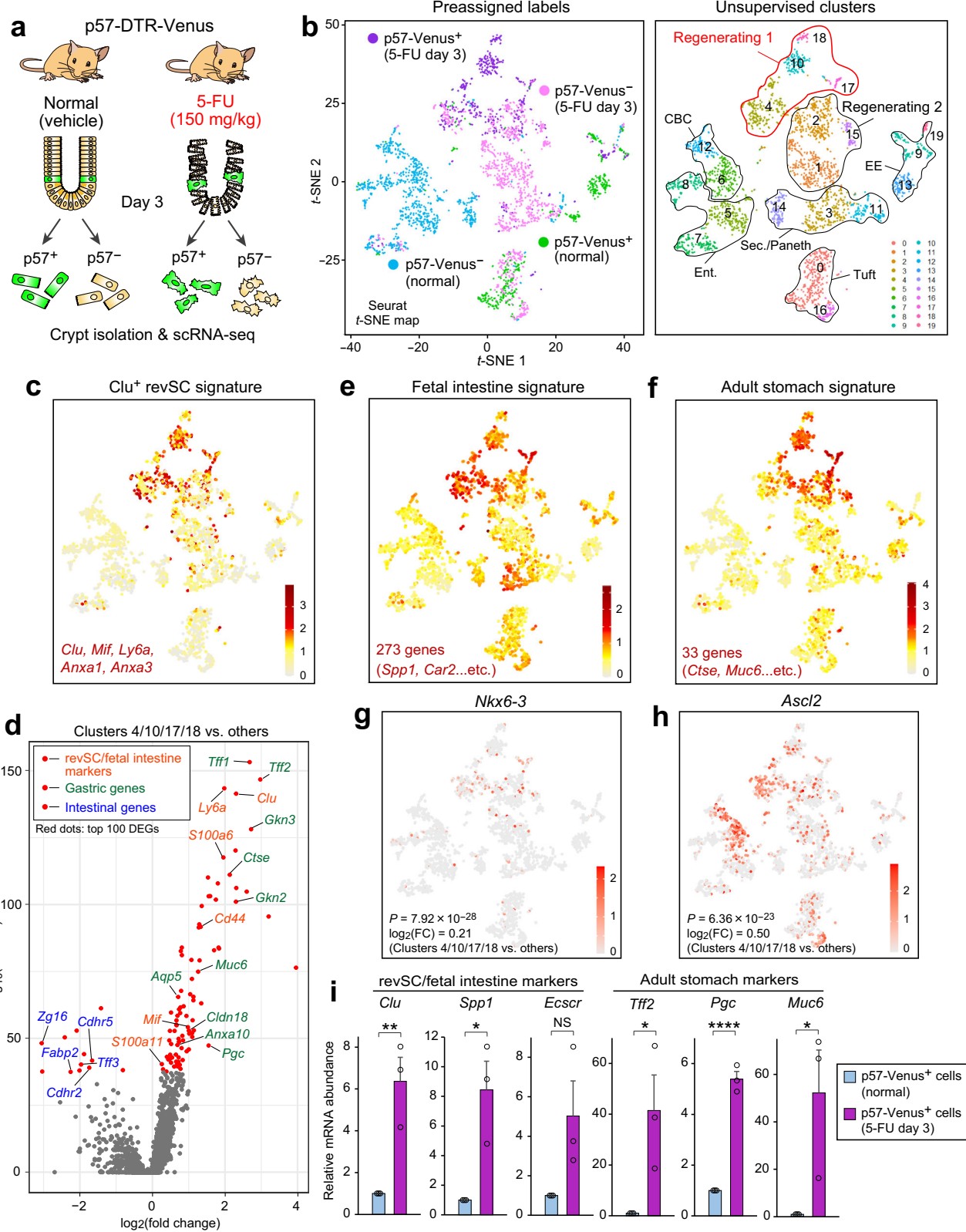

transcriptomic profiles in association with spatiotemporal reprogramming during postinjury regeneration.

**Histological analysis of spatiotemporal reprogramming.** To investigate whether postinjury spatiotemporal reprogramming takes place exclusively in p57[+] cells or also occurs in other cell types, we sorted absorptive enterocyte lineage cells, which have

been shown to constitute a major population of dedifferentiation-compatible progenitors[16], from the crypts of normal or 5-FU-treated mice with the use of the enterocyte-specific cell surface marker Anpep (Supplementary Fig. 6a). RT-qPCR analysis revealed that the expression of genes characteristic of fetal intestine or adult stomach was markedly upregulated in post-injury Anpep[+] cells compared with their normal counterparts

**Fig. 5 Single-cell transcriptomics analysis uncovers injury-induced spatiotemporal reprogramming of p57+ cells. a** Schematic representation of the experimental procedure for scRNA-seq analysis of normal and regenerating crypt cells from the small intestine of p57-DTR-Venus mice ($n = 2$ biological replicates for each condition, with two mice for each replicate). **b** A $t$-SNE projection for 567 p57-Venus+ (purple) and 1055 p57-Venus− (pink) crypt cells from p57-DTR-Venus mice at 3 days after injection of 5-FU as well as for 472 p57-Venus+ (green) and 923 p57-Venus− (cyan) crypt cells from normal p57-DTR-Venus mice is shown in the left panel. Unsupervised clusters are color coded and overlaid on the $t$-SNE map in the right panel. The indicated cellular identity of each cluster is based on the expression of known cell-type-specific genes (Supplementary Fig. 4d, e). **c** Heat map for the mean expression level of the indicated Clu+ revSC marker genes shown on the $t$-SNE projection. **d** Volcano plot showing $\log_2$(fold change) and $-\log_{10}(P$ value) for differentially expressed genes (DEGs) in the Regenerating 1 group (clusters 4, 10, 17, and 18) relative to all other cell clusters. Points colored red indicate the top 100 genes with regard to $-\log_{10}(P$ value). Typical revSC- or fetal intestine-specific genes (orange), adult stomach-specific genes (green), and adult intestine-specific genes (blue) are highlighted. **e, f** Heat maps for the mean expression level of fetal intestine signature genes (**e**) or of adult stomach signature genes (**f**) shown on the $t$-SNE projection. **g, h** Heat maps for the abundance of *Nkx6-3* (**g**) and *Ascl2* (**h**) mRNAs shown on the $t$-SNE projection. FC, fold change. **i** RT-qPCR analysis of relative mRNA abundance for representative marker genes of revSCs or fetal intestine, or for those of adult stomach, in sorted normal and postinjury p57-Venus+ cells. Data are means + SEM ($n = 3$ mice). NS (not significant), *$P < 0.05$, **$P < 0.01$, ****$P < 0.001$ (two-tailed Student's $t$ test). Individual genes for each signature are listed in Supplementary Table 1. Source data are provided as a Source Data file.

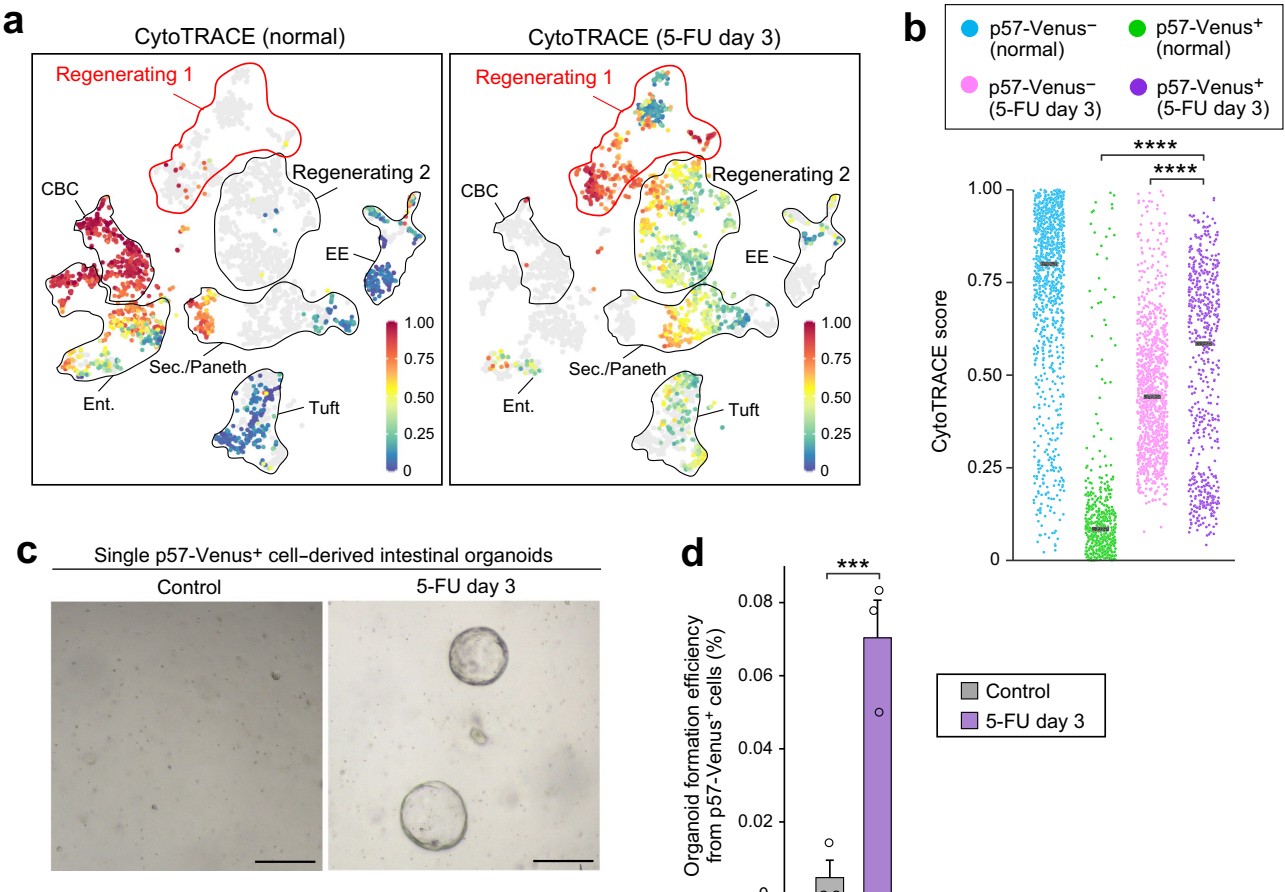

**Fig. 6 p57+ cells acquire a stem cell-like transcriptomic profile in association with spatiotemporal reprogramming. a** CytoTRACE analysis of our scRNA-seq data. The higher the score, the greater the stem cell potential. The calculated CytoTRACE scores for normal or postinjury crypt cells are overlaid separately on the $t$-SNE projection shown in Fig. 4b. **b** CytoTRACE score for each cell. Cells are grouped and color coded on the basis of p57-Venus expression status and experimental condition ($n = 567$ postinjury p57-Venus+ cells, 1055 postinjury p57-Venus− cells, 472 normal p57-Venus+ cells, and 923 normal p57-Venus− cells from 2 biologically independent replicates). Bars indicate median values. ****$P < 0.001$ (Steel-Dwass test). **c** Representative phase-contrast images of sorted p57-Venus+ cell-derived organoids at 8 days after plating. Single cells were isolated from p57-DTR-Venus mice at 3 days after intraperitoneal injection of vehicle (left) or 5-FU at 150 mg/kg (right). Scale bars, 100 μm. **d** Efficiency of organoid formation from sorted single p57-Venus+ cells isolated from p57-DTR-Venus mice as determined at 8 days after plating as in **c**. Data are means + SEM ($n = 3$ mice). ***$P < 0.005$ (two-tailed Student's $t$ test). Source data are provided as a Source Data file.

(Fig. 7a), suggesting that spatiotemporal reprogramming is indeed operative not only in p57+ EE/tuft lineage cells but also in other cell types during postinjury regeneration.

We next examined whether spatiotemporal reprogramming is detectable at the histological level. In addition to the typical acute injury model based on a single injection of 5-FU at 150 mg/kg (SF

protocol), we examined the effects of the administration of both 5-FU and probenecid (FP protocol) (Fig. 7b), the latter of which inhibits the renal excretion of multiple drugs[40]. The FP protocol resulted in increased morbidity of mice, which was likely attributable to the prolonged retention of 5-FU or its metabolic products in blood. Most mice treated with the FP protocol thus

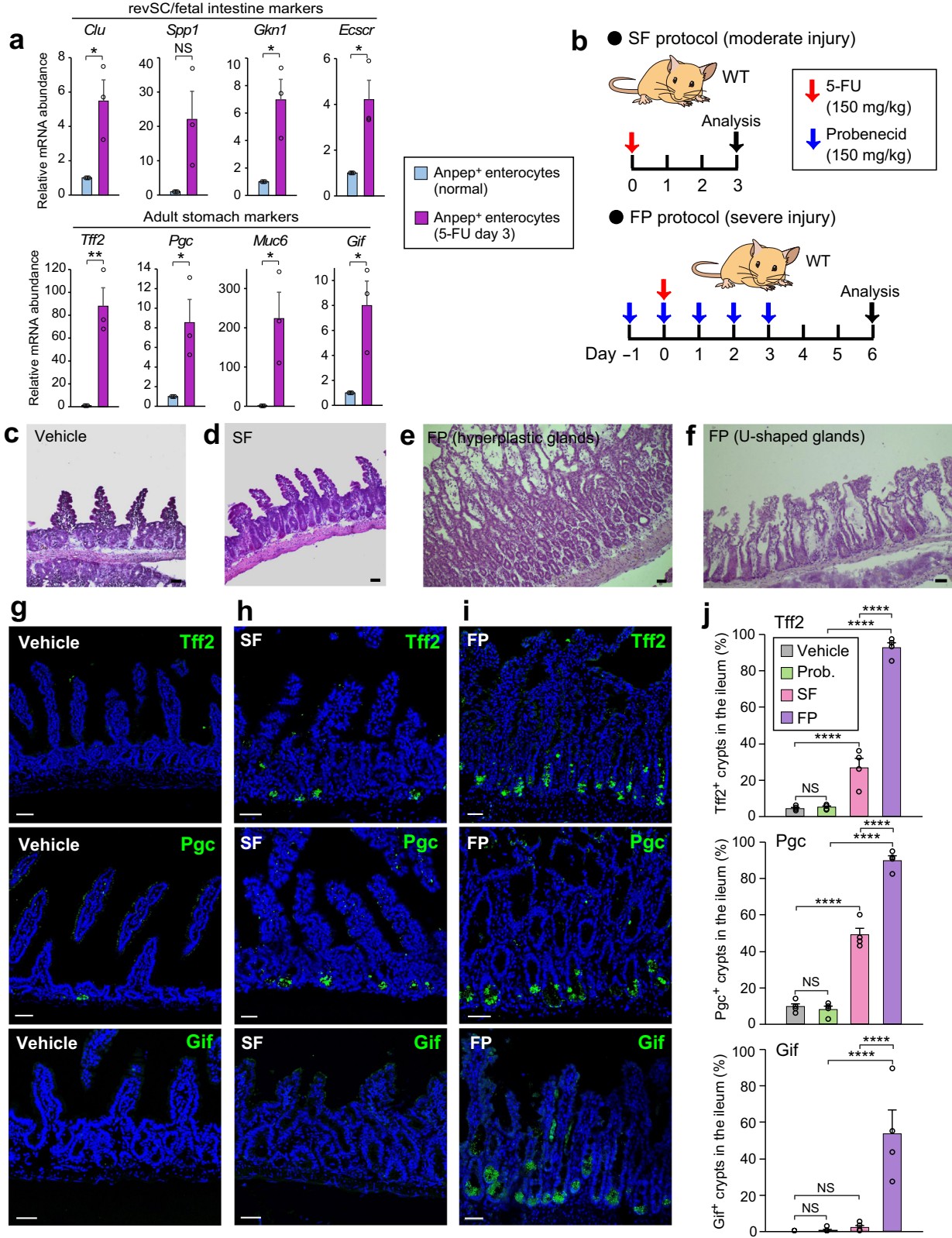

manifested sustained weight loss and died within 6 to 10 days after 5-FU injection, whereas the remainder that had regained body weight by days 2 to 4 survived (Supplementary Fig. 6b, c). Although we did not observe any substantial morphological changes in the intestine of SF-treated mice compared with vehicle-treated control animals (Fig. 7c, d), the ileum of FP-treated mice with sustained weight loss manifested an aberrant

epithelial morphology with marked crypt hyperplasia and the loss of apparent crypt-villus structures (Fig. 7e), characteristics of severe intestinal injury. We hereafter applied the SF and FP protocols as models for moderate and severe intestinal injury, respectively. We also found that the morphologically aberrant regions of the intestine of FP-treated mice included a number of U-shaped glands (Fig. 7f), a finding reminiscent of the pyloric

**Fig. 7 Characterization of spatiotemporal reprogramming in the intestinal epithelium at the population and histological levels. a** RT-qPCR analysis of relative mRNA abundance for representative marker genes of revSCs or fetal intestine or for those of adult stomach in sorted Anpep+ enterocytes from control or 5-FU–treated WT mice. Data are means + SEM ($n = 3$ mice). NS (not significant), *$P < 0.05$, **$P < 0.01$ (two-tailed Student's $t$ test). **b** Dosing regimen for WT mice subjected to moderate (SF protocol) or severe (FP protocol) intestinal injury. **c–f** Representative hematoxylin-eosin staining for the ileum of mice treated with vehicle (phosphate-buffered saline containing 10% dimethyl sulfoxide) (**c**), the SF protocol (**d**), or the FP protocol (**e, f**). **g–i** Representative immunofluorescence staining of Tff2, Pgc, or Gif in the ileum of mice treated with vehicle (**g**), the SF protocol (**h**), or the FP protocol (**i**). **j** Quantification of the fraction of Tff2+, Pgc+, or Gif+ crypts in the ileum of mice treated with vehicle, with probenecid alone (Prob., images not shown), with the SF protocol, or with the FP protocol determined from images as in **g** to **i**. The microscopic fields were randomly selected, and a crypt containing at least three marker-positive cells was regarded as a marker-positive crypt. Four mice and 200 crypts per mouse were analyzed for each condition. Data are means + SEM. NS (not significant), ****$P < 0.001$ (Tukey-Kramer test). Scale bars, 50 μm. Source data are provided as a Source Data file.

metaplasia apparent in Crohn's disease[22]. These results suggested that severe intestinal injury induces morphological changes that resemble gastric metaplasia.

We next performed immunofluorescence staining of several markers for fetal intestine and adult stomach in the normal or injured small intestinal epithelium. The small intestine of SF-treated mice manifested positivity for Sca1 staining in crypts (Supplementary Fig. 6d), increased mucosal neutrophil infiltration (Supplementary Fig. 7a–c), and up-regulation of interferon-γ gene (Ifng) expression in several lymphocyte (T cell, natural killer [NK] cell, and innate lymphoid cell [ILC] 1/2) populations (Supplementary Fig. 7d, e), indicative of the occurrence of fetal-like reversion[19,20]. In addition, we found that Pgc and Tff2— markers for gastric chief and mucous cells, respectively—were expressed in crypts of the jejunum (Supplementary Fig. 6e–h) and ileum (Fig. 7g–j) of SF-treated mice at 3 days after 5-FU injection, with such expression being almost or completely undetectable in control mice treated with vehicle or with probenecid alone. Furthermore, the expression of these gastric markers was even more significantly upregulated in the FP-treated mice, consistent with the pronounced morphological changes apparent in these animals. The crypts of FP-treated mice were also positive for Gif, another marker for chief cells, with such staining not being apparent in control or SF-treated mice. The up-regulation of gastric cell markers was also apparent at both histological (Supplementary Fig. 8a–e) and gene expression (Supplementary Fig. 8f) levels in the case of IR-induced injury, suggesting that this process is not specific to 5-FU treatment but is common to intestinal injury contexts. Collectively, these results suggested that spatiotemporal reprogramming proceeds universally in multiple cell types, rather than specifically in p57+ cells, of postinjury crypts in proportion to the extent of the induced damage.

**p57+ cells and spatiotemporal reprogramming in adenoma.** Cancer stem cells manifest marked plasticity[41], which is often analogous to that of normal tissue stem cells during tissue regeneration[42,43]. Recent studies have indicated that Lgr5− cells play key roles in the maintenance[44] and metastasis[45] of intestinal cancers. Despite previous studies showing injury-inducible stem cell activity of EE lineage cells in normal intestinal crypts[13,14], the dynamics of such cells in the neoplastic context have not been directly characterized. We therefore finally investigated the stem cell activity of p57+ cells in adenomas of the small intestine. Immunofluorescence staining of intestinal tumors from Apc$^{\Delta716}$ mice[46] revealed that ~1% of adenoma cells expressed p57 (Fig. 8a). RT-qPCR analysis revealed preferential expression of marker genes for EE cells or intestinal LRCs in p57-Venus+ cells versus total adenoma cells from Apc$^{\Delta716}$/p57-DTR-Venus mice (Fig. 8b, c). Tracing experiments with Apc$^{\Delta716}$/p57-CreER$^{T2}$/CAG-LSL-tdTomato mice (Fig. 8d) showed that the initially labeled cells after injection of tamoxifen were completely negative for Ki67 staining (Fig. 8e). These results thus suggested that, similar to their normal counterparts, p57+ cells in intestinal

adenoma remain quiescent and retain EE cell identity. In contrast to the complete absence of stem cell activity for p57+ normal crypt cells in the steady-state (Fig. 4), however, we observed lineage tracing events for p57+ adenoma cells (Fig. 8f), with the width of segments ranging from 50 to 500 μm and at least one tracing event being apparent in ~17% of microscopically visible tumors (mean ± SEM = 17.2 ± 1.01%, $n = 3$ mice). These clones were readily detectable 2 weeks after tamoxifen injection, typically emanated from the base and progressed to the top of the glandular structures, and contained numerous Ki67+ actively cycling adenoma cells (Fig. 8g) and, importantly, Lgr5+ conventional adenoma stem cells[44] (Fig. 8h). These findings suggested that p57+ EE-like cells in adenoma constitutively revert to the stem cell state and give rise to heterogeneous adenoma cell populations.

Consistent with these findings, immunofluorescence analysis revealed that the fetal intestine marker Sca1 (Fig. 8i) and the adult gastric markers Tff2 (Fig. 8j), Pgc (Fig. 8k), and Gif (Fig. 8l) were robustly expressed in multiple cells in a subset of glandular structures of adenoma (mean ± SEM = 22.4 ± 3.49%, 27.7 ± 4.72%, 19.4 ± 3.92%, and 7.17 ± 2.21% of glands, respectively, $n = 3$ mice), similar to our findings for spatiotemporal reprogramming in regenerating normal crypts (Fig. 7). Together, these results indicated that, in intestinal adenoma, a differentiated cell lineage spontaneously acquires stem cell-like capacity even under homeostatic conditions, in the context of constitutive activation of spatiotemporal reprogramming.

**Discussion**

We have here shown that p57 specifically marks an EE/tuft cell progenitor that is located around the +4 position of intestinal crypts and which serves as a facultative ISC after epithelial injury. Characterization of the dedifferentiation of these p57+ cells has provided insight into the process by which lineage-committed intestinal crypt cells revert to the ISC state and regenerate the damaged epithelium.

The lack of a specific marker for so-called "quiescent +4 ISCs" has led to a long-standing debate as to the existence of a dedicated, undifferentiated type of quiescent ISC[11]. Although p57 expression is highly specific for the rare quiescent cells around the +4 position of crypts, our tracing experiments have shown that these cells are EE/tuft lineage-committed cells in the steady-state but that they acquire stem cell potential through a dedifferentiation process after crypt injury. Together with previous tracing studies of LRCs[13] or other EE/tuft lineage cells[14], our results support the notion that quiescent +4 ISCs might correspond to the dedifferentiation-compatible EE/tuft cell progenitors, although we are unable to formally exclude the possible existence of unidentified, primitive quiescent ISCs in intestinal crypts.

Our scRNA-seq analysis revealed that the p57+ EE/tuft lineage cells undergo a dynamic reprogramming process associated with their postinjury dedifferentiation. This process is characterized by concomitant fetal-like reversion and gastric-like conversion of

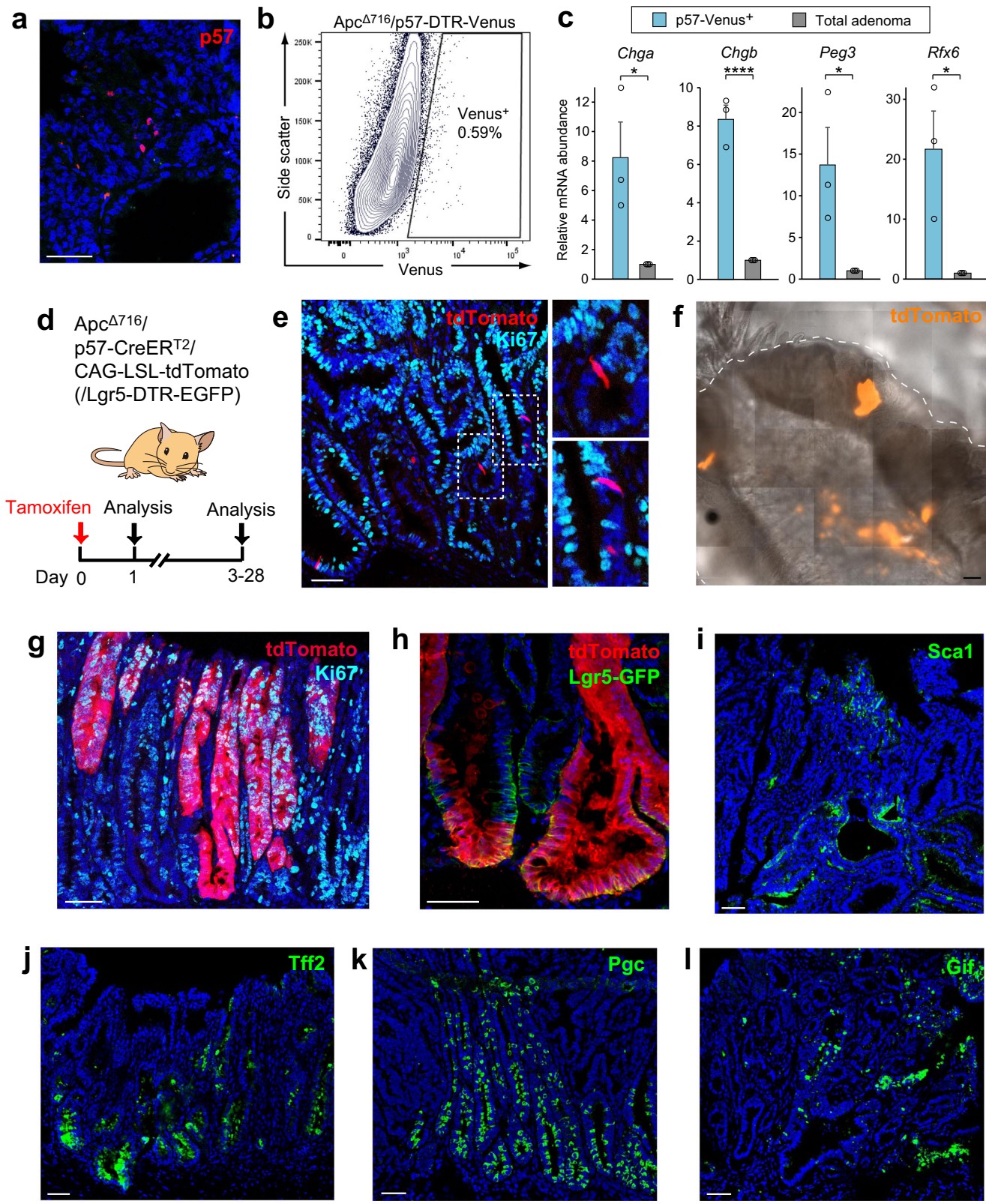

transcriptomic signatures that allow the p57[+] cells to serve as a preexisting source of intermediary ISCs[18–20] that support subsequent epithelial regeneration. Further histological and population-level analyses showed that such changes in gene expression occur throughout the intestinal crypt after injury. Given recent studies showing that multiple differentiated cell types are capable of reversion to the ISC state, it is possible that such spatiotemporal reprogramming is a universal process that accompanies the dedifferentiation of committed cell types, rather than a phenomenon restricted to one cell population (such as p57[+] cells). This notion suggests that exploration of regulatory factors for spatiotemporal reprogramming might lead directly to the elucidation of the prevailing mechanisms underlying the dedifferentiation of any intestinal cell type.

**Fig. 8 p57$^+$ EE lineage cells in intestinal adenoma manifest homeostatic stem cell activity in the context of constitutively activated spatiotemporal reprogramming. a** Representative immunofluorescence staining of p57 in intestinal adenoma of an Apc$^{\Delta716}$ mouse. **b** Representative FACS plot for adenoma cells from Apc$^{\Delta716}$/p57-DTR-Venus mice and gating strategy for isolation of p57$^+$ cells. **c** RT-qPCR analysis of relative mRNA abundance for EE cell markers (*Chga* and *Chgb*) and LRC markers (*Peg3* and *Rfx6*) in FACS-sorted p57-Venus$^+$ and total adenoma cells as in **b**. Data are means + SEM ($n = 3$ mice). *$P < 0.05$, ****$P < 0.001$ (two-tailed Student's $t$ test). **d** Schematic representation of labeling or lineage tracing for p57$^+$ cells in intestinal adenoma of mice harboring the Apc$^{\Delta716}$ allele. **e** Confocal immunofluorescence analysis of Ki67 expression and tdTomato-labeled p57$^+$ cells in intestinal adenoma at 24 h after tamoxifen injection in an Apc$^{\Delta716}$/p57-CreER$^{T2}$/CAG-LSL-tdTomato mouse. The boxed regions of the left panel are shown at higher magnification on the right. No Ki67$^+$tdTomato$^+$ cells were apparent. **f** Representative image (tdTomato fluorescence and differential interference contrast) of a cleared whole-mount intestinal adenoma preparation from an Apc$^{\Delta716}$/p57-CreER$^{T2}$/CAG-LSL-tdTomato mouse at 1 month after tamoxifen injection. The dashed line indicates the tumor outline. **g, h** Confocal immunofluorescence images for the tdTomato-labeled p57$^+$ cell-derived lineage stained for Ki67 (**g**) or for Lgr5-GFP (**h**) in intestinal adenoma of Apc$^{\Delta716}$/p57-CreER$^{T2}$/CAG-LSL-tdTomato or Apc$^{\Delta716}$/p57-CreER$^{T2}$/CAG-LSL-tdTomato/Lgr5-DTR-EGFP mice, respectively, at 1 month after tamoxifen injection. **i–l** Representative immunofluorescence staining of the fetal intestinal marker Sca1 (**i**) and the adult stomach markers Tff2 (**j**), Pgc (**k**), and Gif (**l**) in intestinal adenoma of Apc$^{\Delta716}$ mice. Scale bars: 50 μm (**a, e, g–l**) or 200 μm (**f**). Source data are provided as a Source Data file.

Although metaplasia is frequently observed clinically in ulcerative or inflammatory conditions that affect the intestinal epithelium, the mechanism underlying this phenomenon has remained poorly understood. We have now shown that 5-FU-induced severe and relatively acute epithelial injury is also accompanied by metaplasia-like molecular and morphological changes, indicating that such metaplastic transformation is driven by a universal postinjury response, rather than being restricted to specific pathologic contexts such as inflammation. Although spatiotemporal reprogramming itself may essentially be a physiological system to replenish the functional ISC pool after regular toxic insults, it is also possible that defects in this process may result in the production of cells with a different histological identity and thereby give rise to metaplasia. In addition, some aspects of the observed metaplasia were also apparent in intestinal adenoma, suggesting that such reprogramming might underlie the marked plasticity of cancer stem cell systems, including the spontaneous acquisition of stemness or lineage infidelity[43] in differentiated cell types. This notion is consistent with the latest study showing that gastric-like metaplasia underlies the initiation of human colorectal cancers[47]. Future studies are warranted to uncover whether similar mechanisms are operative in other epithelial or nonepithelial systems.

In summary, our present results shed light on the plasticity of systems for maintaining tissue integrity in the normal and neoplastic intestinal epithelium. Elucidation and manipulation of the molecular mechanisms underlying such plasticity are likely to be of clinical importance in the field of regenerative medicine as well as in cancer therapy. Future studies on p57$^+$ cells should provide further insight into the diversity of systems underlying tissue maintenance as well as contribute to the identification of unexplored stem cell populations in various tissue and disease contexts.

## Methods

The research performed in this paper complies with all relevant ethical regulations. All mouse experiments were approved by the Animal Ethics Committee of Kyushu University. All experiments that require genetic manipulation were approved by the Safety Committee for Recombinant DNA experiments of Kyushu University.

**Animal models.** p57-DTR-Venus and p57-CreER$^{T2}$ mice were generated by gene targeting in mouse embryonic stem (ES) cells and the subsequent establishment of chimeric mice (see below). Generation of p57$^{+/F}$ mice was described previously[23]. Lgr5-EGFP-IRES-CreER$^{T2}$ mice[1] and CAG-LSL-tdTomato mice[27] were obtained from The Jackson Laboratory. Lgr5-DTR-EGFP mice[3] were kindly provided by F. de Sauvage. All mice harboring genetic modifications of the p57 gene locus were generated in a manner that ensured the modifications were maternally inherited. For lineage tracing of p57$^+$ cells, female mice harboring the p57-CreER$^{T2}$ construct were crossed with male mice harboring the CAG-LSL-tdTomato cassette. The resulting offspring were also crossed with mice harboring the Lgr5-DTR-EGFP or Apc$^{\Delta716}$ (ref. [46]) constructs as indicated. For intestine-specific inducible knockout

of p57 and visualization of Bmi1$^{high}$ + 4 position cells, female p57$^{+/F}$ mice were crossed with male mice harboring the Villin-CreER$^{T2}$ (ref. [25]) and Bmi1-GFP[48] constructs. All mice were housed in the specific pathogen-free animal facility at Kyushu University in accordance with institutional guidelines under the following conditions: 22 °C ambient temperature, 50–60% humidity, 12 h dark/light cycle, and free access to water and standard rodent chow. Animal experiments were performed with 8- to 18-week-old male and female mice on the C57BL/6 × 129/Sv background.

**Generation of p57 knock-in mice.** Both p57-DTR-Venus and p57-CreER$^{T2}$ mice were generated as previously described[23], with a slight modification in that the clustered regularly interspersed short palindromic repeats (CRISPR)–Cas9 system was applied to facilitate homologous recombination. In brief, the knock-in constructs were designed so that T2A-DTR-T2A-Venus or T2A-CreER$^{T2}$ cassettes were inserted in-frame into the endogenous p57 coding sequence immediately 5' to the only TGA codon at this locus. The 5' and 3' regions of homology were generated by genomic PCR with appropriate primers, and each final construct was cloned into the multiple cloning site of pBluescript SK II (+). Embryonic day–4 ES cells were cultured on a feeder layer of mouse embryonic fibroblasts in Dulbecco's modified Eagle's medium (DMEM) supplemented with fetal bovine serum (FBS) and leukemia inhibitory factor. They were electroporated with the targeting vector, the bicistronic CRISPR vector pX330 (Addgene Cat# 42230) encoding Cas9 and a p57-specific single guide RNA (5'-GGCUCCUCCGGAUGGGCUCA-3'), and the pPGK-Puro vector (Addgene Cat# 11349). The transfected cells were selected for 36 h in the presence of puromycin (1 μg/ml), expanded for 7 days, and picked up as single colonies. ES cell clones that had undergone homologous recombination were identified by PCR and confirmed by Southern blot analysis. They were then expanded and injected into C57BL/6 mouse blastocysts to generate chimeric mice. The germline transmission of the p57-DTR-Venus or p57-CreER$^{T2}$ allele was achieved by crossing of chimeras with C57BL/6 mice. Primer sequences for vector construction are available on request.

**Mouse treatments.** For lineage-tracing experiments, mice were injected intraperitoneally with a single dose (50 mg/kg) of tamoxifen (Sigma Cat# T5648) to activate Cre-induced tdTomato expression. Inducible knockout of p57 was performed by intraperitoneal injection of tamoxifen (50 mg/kg) on each of 5 consecutive days. For induction of crypt injury, mice were injected intraperitoneally with 5-FU (Sigma Cat# F6627) at 150 mg/kg or subjected to whole-body irradiation with 6, 8.5, or 12 Gy of IR, as indicated in figure legends. For in vivo activation of Wnt signaling, CHIR99021 (Axon Medichem Cat# 1386) was injected intraperitoneally at a dose of 1 mg/kg. For induction of severe crypt injury in Figs. 6, 5-FU treatment was combined with intraperitoneal injection of probenecid at 150 mg/kg per day for 5 consecutive days. Ablation of p57$^+$ or Lgr5-expressing cells with the use of p57-DTR-Venus or Lgr5-DTR-EGFP constructs, respectively, was performed by intraperitoneal injection of DT (Millipore Cat# 322336) at 50 μg/kg. For the label retention assay, BrdU (Sigma Cat# 5002) at a dose of 40 mg/kg was injected intraperitoneally three times per day on 3 consecutive days and the analysis was performed 10 days after the last injection. For detection of cells in S phase, EdU (Life Technologies Cat# A10044) at a dose of 20 mg/kg was injected intraperitoneally three times at 2-h intervals and the analysis was performed 6 h after the first injection.

**Histological analysis.** The intestine was flushed with 4% paraformaldehyde in phosphate-buffered saline (PBS), opened longitudinally, and fixed overnight at 4 °C with 4% paraformaldehyde in PBS. For immunofluorescence analysis, the fixed and cryoprotected tissue was embedded in OCT compound (Tissue Tek Cat# 4583), frozen, and cut into sections at a thickness of 4 to 10 μm with a cryostat. Dried sections were rehydrated overnight at 4 °C in PBS and subjected to immunostaining. For staining of BrdU, Lyz1, Pgc or Gif, sections were also subjected to

antigen retrieval in 10 mM citrate buffer for 10 min at 95 °C. All sections were incubated overnight at room temperature with primary antibodies diluted in PBS containing 2% bovine serum albumin and 0.3% Triton X-100. The following primary antibodies were used at indicated dilutions: Goat polyclonal anti-p57 (M-20) (Santa Cruz Cat# sc-1039) at a 1:20 dilution; Rabbit polyclonal anti-p57 (H-91) (Santa Cruz Cat# sc-8298), Rabbit monoclonal anti-Ki67 (SP6) (Thermo Fisher Scientific Cat#MA5-14520), Biotinylated anti-5-bromo-2'-deoxyuridine (BrdU) (BD Biosciences Cat# 550803), Goat polyclonal anti-Chga (C-20) (Santa Cruz Cat#sc-1488), Goat polyclonal anti-Lyz1 (C-19) (Santa Cruz Cat#sc-27958), Rabbit polyclonal anti-Tff2 (Proteintech Cat#13681-1-AP), Rabbit polyclonal anti-Pgc (Bioss Cat#bs-1551R), and Rabbit polyclonal anti-Gif (Cloud-Clone Cat#-PAD111Mu01) at a 1:100 dilution; Chicken polyclonal anti-GFP (Aves Cat#GFP1020), Rabbit polyclonal anti-RFP (Abcam Cat#ab62341), Rabbit polyclonal anti-Dclk1 (Abcam Cat#ab31704), Goat polyclonal anti-Anpep (R&D Systems Cat#AF2335), and Rabbit polyclonal anti-Muc2 (Novus Cat#NBP1-31231) at a 1:100 dilution. The sections were then washed four times with PBS before incubation for 3 h at room temperature with species-appropriate Alexa Fluor 546–, Alexa Fluor 488–, or fluorescein isothiocyanate-conjugated secondary antibodies (Thermo Fisher Scientific Cat# A10040, Cat# A11008, or Cat# A11055; Aves Cat# F1005) for detection of unconjugated primary antibodies or with Alexa Fluor 488-conjugated streptavidin (Thermo Fisher Scientific Cat# S11223) for detection of biotin-conjugated primary antibodies. All secondary antibodies were used at a 1:500 dilution. Staining for EdU was performed with the use of a Click-iT EdU Alexa Fluor 555 Imaging Kit. After staining was completed, the sections were washed four times and then mounted with the use of Vectashield mounting medium containing 4',6-diamidino-2-phenylindole (DAPI) (Vector Laboratories Cat# H-1200). Nuclei were thus stained with DAPI and displayed with blue in all immunofluorescence images. Imaging and quantification were performed with an LSM700 confocal microscopy system (Zeiss).

**Whole-mount observation of lineage tracing events.** Tissue clearing was performed according to the SeeDB protocol[49]. In brief, the fixed intestine was sequentially immersed in PBS containing 20, 40, 60, or 80% fructose (Sigma Cat# F0127) for 4 to 8 h at each step and with gentle rotation at room temperature. The tissue was then incubated in 100% fructose in water for 24 h at room temperature and embedded in the same solution. Screening, counting, and imaging of lineage tracing events were performed with a DeltaVision Elite microscopy system (GE Healthcare Life Sciences).

**Isolation of single crypt cells, adenoma cells, and immune cells from the intestine.** For isolation of single crypt cells from the normal epithelium, the intestine was flushed with ice-cold sterile PBS, opened longitudinally, and scraped with a sterile glass microscope slide to remove villi. The tissue was cut into 5-mm pieces, passed repeatedly through the tip of a pipette in sterile PBS at 4 °C, and incubated in PBS containing 2 mM EDTA for 30 min at 4 °C. Crypts were isolated by vigorous pipetting of the intestinal pieces, the passage of the suspension through a 70-μm cell strainer, and centrifugation of the filtrate at 300 × g for 5 min at 4 °C. The crypt pellet was incubated for 30 min at 37 °C in TrypLE Express (Life Technologies Cat# 12605) containing DNase (Sigma Cat# DN25) at 100 μg/ml and was dissociated into single cells by intermittent pipetting. The cells collected by centrifugation at 300 × g for 5 min at 4 °C and washed with Advanced DMEM/F12 (Sigma Cat# 12634).

Single adenoma cells were isolated as described previously[50]. In brief, tumors arising in the small intestine were visually isolated, minced, and incubated for 30 min at 37 °C in DMEM supplemented with 10% FBS, penicillin-streptomycin (Invitrogen Cat# 15070-063), collagenase (Wako Cat# 034-22363) at 2 mg/ml, and DNase (100 μg/ml). The cells were isolated by vigorous pipetting of the tissue pieces, passage through a 70-μm cell strainer, and incubation for 30 min at 4 °C in PBS containing 2 mM EDTA with intermittent pipetting.

For isolation of resident or infiltrated immune cells of intestinal epithelial and submucosal tissue, 5-mm pieces of small intestine without villi were prepared as described above and incubated for 30 min at 37 °C in DMEM supplemented with 10% FBS, penicillin-streptomycin, and collagenase at 2 mg/ml. The cells were isolated by vigorous pipetting of the tissue pieces, passage through a 70-μm cell strainer, and incubation for 10 min at 37 °C in TrypLE Express containing DNase at 100 μg/ml.

**FACS and flow cytometric analysis.** Cell sorting and analysis were performed with a FACS Aria IIu instrument (BD Biosciences) fitted with a 100-μm nozzle, and the data were analyzed by FlowJo_V10 software. For sorting or analysis of live cells, isolated crypt cells were stained first for 30 min at 4 °C with biotin-conjugated monoclonal antibodies to EpCAM (G8.8) (eBiosciences Cat# 13-5791-82) at a 1:500 dilution in FACS buffer (sterile PBS containing 4% FBS) and then with streptavidin conjugated with an appropriate fluorophore. For analysis and sorting of Anpep+ enterocytes, the crypt cells were stained first for 30 min at 4 °C with polyclonal antibodies to Aminopeptidase N/CD13 (R&D systems Cat# AF2335) at a 1:500 dilution in FACS buffer and then with antibodies to Goat IgG conjugated with Alexa Fluor 488 (Thermo Fisher Scientific Cat# A11055) at a 1:500 dilution in FACS buffer. The cells were washed and then resuspended in FACS buffer

containing PI (2 μg/ml). After gating for forward scatter and side scatter and exclusion of doublets, the PI−EpCAM+ fraction was collected as single live crypt cells, and each stem cell population was defined on the basis of the fluorescence intensity of each reporter.

For cell cycle analysis by Ki67/PI staining, isolated crypt cells (1 × 10^6) were first stained for 30 min at 4 °C in ice-cold PBS with the use of a LIVE/DEAD Fixable Near-IR Cell Stain Kit (Invitrogen Cat# L10119) at a 1:500 dilution, and were then subjected to EpCAM staining as described above. They were then fixed for 15 min at room temperature with 4% paraformaldehyde in PBS before permeabilization by incubation for 15 min at 4 °C in PBS containing 0.1% Triton X-100. Ki67 staining was performed by incubation of the cells for 30 min at 4 °C with FACS buffer containing 0.1% Triton X-100 and Alexa Fluor 647-conjugated monoclonal antibodies to Ki67 (16A8) (BioLegend Cat# 652407) at a 1:200 dilution. The cells were then diluted in PBS containing PI (1 μg/ml) for analysis. Cells were washed with FACS buffer or PBS once after each treatment or antibody staining.

For analysis of neutrophils, cells isolated from the small intestine were stained for 30 min at 4 °C with phycoerythrin- and Cy7-conjugated monoclonal antibodies to CD45 (104) (BioLegend Cat# 109829), Alexa Fluor 488-conjugated monoclonal antibodies to Gr1 (RB6-8C5) (BioLegend Cat# 108417), and allophycocyanin-conjugated monoclonal antibodies to CD11b (M1/70) (BioLegend Cat# 101211). For sorting of lymphocytes, cells isolated from the small intestine were stained for 30 min at 4 °C with phycoerythrin- and Cy7-conjugated antibodies to CD45, allophycocyanin- and Cy7-conjugated monoclonal antibodies to T cell receptor (TCR) β (H57-597) (BioLegend Cat# 109219), biotin-conjugated monoclonal antibodies to TCRγδ (GL3) (BioLegend Cat# 118103), phycoerythrin-conjugated monoclonal antibodies to NK1.1 (PK136) (eBioscience Cat# 12-5941-81), and fluorescein isothiocyanate-conjugated monoclonal antibodies to CD90.2 (53-2.1) (BioLegend Cat# 140303) and then with allophycocyanin- and Cy7-conjugated streptavidin (BD Biosciences Cat# 554063). All antibodies were diluted 1:200 in FACS buffer. The cells were washed and resuspended in FACS buffer containing PI (2 μg/ml) and analyzed as shown in Supplementary Figure 7.

**Organoid culture.** The intestinal organoid culture was performed as previously described[28]. Briefly, the organoids were embedded in 30 μl Matrigel™ (BD Biosciences Cat# 356231), and maintained in 250 μl Advanced DMEM/F12 supplemented with Glutamax (Invitrogen Cat# 35050), B27™ (Thermo Fisher Scientific Cat# 17504), N-2 (Thermo Fisher Scientific Cat# 17502), 50 ng/ml EGF (Thermo Fisher Scientific Cat# PMG8043), 1 μg/ml R-spondin 1 (R&D Cat# 4645-RS), 100 ng/ml Noggin (Pepro Tech Cat3# 250-38). For organoid formation from single intestinal crypt cells, 10 μM Y-27632 was added to the medium for the first 3 days. For Wnt hyperactivation as shown in Supplementary Fig. 3b–c, the medium was further supplemented with 1 μM CHIR99021 and/or a half volume of Wnt3a-conditioned Advanced DMEM/F12 medium (produced using the L Wnt-3A ATCC® CRL-2647™ cell line) in indicated experimental conditions. For in vitro injury or p57+ cell ablation experiments as shown in Fig. 4l–n, organoids derived from the small intestine of p57-DTR-Venus mice were cultured in ENR medium in the absence or presence of 1 μg/ml 5-FU or 5 ng/ml DT according to the time schedule indicated in Fig. 4l, and the efficiency of organoid formation was determined at 14 days after replating. The organoid-forming efficiency was calculated as the percentage of the number of organoids formed at the indicated days to the number of plated cells.

**RT-qPCR analysis.** Cells (500–2000) were directly sorted into a tube containing 800 μl of Isogen (Nippon Gene Cat# 319-90211) for isolation of total RNA, with the use of Gene-Packman Coprecipitant (Nacalai Tesque Cat# 12680-30) to facilitate RNA precipitation. RT was performed with the use of a QuantiTect Reverse Transcription Kit (Qiagen Cat# 205313), and the resulting cDNA was subjected to real-time PCR analysis with SYBR Green PCR Master Mix (Takara Cat# RR420A) and specific primers in a Step One Plus Real-Time PCR System (Applied Biosystems). The abundance of target mRNAs was normalized by that of *Hprt1* mRNA. Primer sequences are listed in Supplementary Table 2.

**scRNA-seq library preparation.** For scRNA-seq analysis of normal or postinjury crypt cells, we injected p57-DTR-Venus mice with vehicle or 5-FU (150 mg/kg) and then isolated the single crypt cells from the duodenum, jejunum, and ileum 3 days later. The cells from two mice were prepared as one sample for each condition and subjected to sequencing analysis (replicate 1). The same experiment was performed at another time with samples from different mice (replicate 2). scRNA-seq library preparation was performed with the use of the CEL-seq2 technique[30]. In brief, each single cell was directly sorted into a well of a 96-well PCR plate containing primer mix including RT primers with unique molecular identifiers (UMIs) and cell barcodes, deoxynucleoside triphosphates, RNase inhibitors, and 0.3% Nonidet P-40. Each library was labeled with the information of experimental condition and FACS gate (for example, p57-Venus+, 5-FU day 3). The primer sequences including cell barcodes were not modified from those in the original CEL-seq2 protocol[30], and are listed in Supplementary Table 3. In total, we sorted 480 p57-Venus+ and 960 p57-Venus− cells from p57-DTR-Venus mice treated with vehicle, and 576 p57-Venus+ and 1152 p57-Venus− cells from p57-DTR-Venus mice treated with 5-FU. The cells were lysed by incubation at 65 °C for

5 min and then subjected to RT with SuperScript III Reverse Transcriptase (Thermo Fischer Cat#18080085). Samples from each plate were pooled into a single library, from which cDNAs were purified with the use of Agencourt AMPure XP Beads (Beckman Coulter Cat# A63381) and subjected to second-strand synthesis. The resultant libraries were again bead-purified and then used as templates for in vitro transcription with the use of a MEGAscript T7 Transcription Kit (Thermo Fischer Cat#AMB13345). The amplified RNAs were subjected to RT followed by PCR amplification (20 cycles) with Illumina TruSeq Small RNA primers to generate the final cDNA libraries. The resultant libraries were subjected to 25-bp (for barcodes) and 75-bp (for gene sequences) paired-end sequencing with Illumina HiSeq2500 (for replicate 1) or NovaSeq6000 (for replicate 2) instruments, so that $\sim 1 \times 10^5$ reads were assigned to each single cell. The sequenced libraries were subjected to bioinformatics analysis as described below.

**Generation of gene-cell matrices for scRNA-seq data**. The FASTQ files obtained from CEL-seq2 analysis were processed through the celseq2 software[30]. The reads were thus "demultiplexed" and mapped to the mouse genome (Mm10, Genome Reference Consortium GRCm38) with the use of Bowtie2 software, and gene-counting was then performed with HTSeq software. The resultant gene (UMI)-cell matrices for each experiment were concatenated with R software. In the case of the data set for intestinal crypt cells from the previous study[14], the sparse matrix data downloaded from GSE99457 were converted to a standard matrix before concatenation.

**Gene expression analysis, dimensionality reduction, and t-SNE visualization**. Visualization of scRNA-seq data was performed mostly with the R package Seurat[26]. For analysis with Seurat v2 of the data set from the previous study[14] shown in Fig. 2, the gene-cell matrix was filtered on the basis of the number of genes detected per cell (cells with values of <200 or >5000 were discarded) and the percentage of mitochondrial UMI counts (cells with a value of >10% were discarded). Variable genes were selected on the basis of their expression and dispersion (expression cutoff = 0.5, dispersion cutoff = 0.5) and were subjected to principal component analysis (PCA) after scaling. t-SNE was run with the first 20 principal components. For analysis of scRNA-seq data of p57-DTR-Venus crypt cells shown in Fig. 4, we used Seurat v3 in combination with Harmony software. Cells with <200 or >5000 genes per cell were discarded. As a result, 567 postinjury p57-Venus⁺ cells, 1055 postinjury p57-Venus⁻ cells, 472 normal p57-Venus⁺ cells, and 923 normal p57-Venus⁻ cells passed the filtering and were subjected to PCA with the use of 2000 identified variable features. The RunHarmony() function was applied to correct for the potential effects of technical differences between sequencing batches (replicates 1 and 2). The RunTSNE() and FindNeighbors() functions were run for clustering, with the use of 15 dimensions identified by Harmony. Genes specifically expressed in each cluster were identified with the FindAllMarkers() function. For the differential gene expression analysis shown in Fig. 4d, g, and h, the FindMarker() function was run for comparison between the Regenerating 1 group (clusters 4, 10, 17, and 18) and all other clusters.

For analysis of the mean expression level of given gene signatures, the normalized and log-scaled expression value of each gene was min-max scaled so that every gene had an equal dynamic range (Max = 1.0, Min = 0), and the mean expression value of the gene signature for each single cell was then calculated. Given that the value of gene expression obtained by Seurat correlates with the absolute quantity of individual mRNAs in each single cell, and in order to reflect this information in the heat maps, the min-max scaled value of signature expression for each cell was finally multiplied by the average maximal expression value of individual genes, so that the resultant dynamic range corresponds to the average absolute amount of the given signature transcripts. The visualization of heat maps on the t-SNE projection was performed with the use of the ggplot2 package in R software.

**CytoTRACE analysis**. For unsupervised exploration of stem cell populations corresponding to scRNA-seq data, the CytoTRACE algorithm[39] was run according to its reference manual with no modification of parameters. The resultant score for each cell was extracted and plotted on t-SNE projection or jitter plot graphs with the use of R software.

**Statistics and reproducibility**. Quantitative data are presented as means ± SEM and were analyzed with the two-tailed Student's t test, the Steel-Dwass test, or the Tukey-Kramer test. The analysis was performed with the use of Excel or R software. A P value of <0.05 was considered statistically significant. For all immunofluorescence analyses without quantification, we confirmed the reproducibility by repetitive experiments on the samples from at least 3 individual mice.

**Reporting summary**. Further information on research design is available in the Nature Research Reporting Summary linked to this article.

## Data availability

The raw data of the scRNA-seq analysis generated in this study have been deposited in DDBJ Sequence Read Archive (DRA) database under accession code DRA013074. The scRNA-seq data from the previous study[14] analyzed in Fig. 2 are available on GSE99457. Source data are provided with this paper.

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

## Acknowledgements
We thank F. de Sauvage (Genentech Inc.) for providing Lgr5-DTR-EGFP transgenic mice, Y. Ohkawa (Medical Institute of Bioregulation, Kyushu University) for discussion on scRNA-seq experiments, M. Ikawa (Research Institute for Microbial Diseases, Osaka, Japan) for discussion on generation of knock-in mice, as well as A. Niihara, H. Takayoshi, M. Tanaka, and K. Nagatoshi for technical assistance. This work was supported in part by KAKENHI grants from Japan Society for the Promotion of Science (JSPS) and the Ministry of Education, Culture, Sports, Science, and Technology of Japan to K.I.N. (JP18H05215) and to T.H. (JP19K16716) as well as by the Project for Cancer Research and Therapeutic Evolution (P-CREATE) of the Japan Agency for Medical Research and Development (AMED) to K.I.N. (JP21cm0106105).

## Author contributions
T.H., Y.O., A.M., S.N., T.O., O.S., D.K., S.T., and H.N. designed and performed experiments, analyzed data, and prepared the manuscript. N.H., S.R., M.M.T., T.S., and K.I.N. contributed to supervision of the study and writing the manuscript.

## Competing interests
The authors declare no competing interests.
