## [Peer Review File · Nature Communications]

Spatiotemporal reprogramming of differentiated cells underlies regeneration and neoplasia in the intestinal epitheliumREVIEWER COMMENTS

Reviewer #1 (Remarks to the Author):

The work contained some high-quality mouse genetic cell lineage analysis and scRNA-seq profiling of p57+ cells in intestinal epithelial plasticity induced by injury. The interpretation of most of the data was appropriate. The notion that gastric metaplastic change may occur not only during injury-induced plasticity but in intestinal tumorigenesis is interesting, although the requirement of p57+ cells for these processes were not substantially explored. Several major issues related to the analysis and interpretation of scRNA-seq data should be addressed.

Major concerns:

1. The authors used p57-CreER mice to demonstrate injury induced lineage tracing in Fig. 3, but used p57-Venus mice for scRNA-seq analysis of injury induced plasticity. As the Venus is controlled under the endogenous p57 promoter (although it can be self-cleaved), and injury turned endogenous p57 gene off (stated by authors on line 242), the scRNA results are expected to be somewhat confounded by such injury-induced alteration of p57 promoter thereby Venus+ cells. It was not shown whether p57-Venus mice also formed ribbons after injury, as p57-CreER mice.
2. Fig. 4b was a little confusing as well as surprising. These scRNA-seq were from crypt cells. The Tuft and EE cells appeared to be great over-represented in tSNE plots of normal crypts than expected. In contrast, the number of Paneth cells, which are major crypt cells, seemed surprisingly under-represented in the tSNE of normal crypts. By comparing the two panels in 4b, it almost indicated that virtually all Paneth cells were only present in 3-day 5-FU treated mice, and they were absent from normal crypts. In addition, authors stated that p57-Venus highlights 1 cell per 20 crypts (line 186). If this was the case, then the proportion of p57-venus-positive vs p57-venus-negative cells represented in these tSNE plots in normal crypts was problematic. These discrepancies raise questions about how the scRNA data were analyzed, and whether the analysis removed substantial numbers of cells from the tSNE plots.
3. Conceptually, on one hand, the authors tried to propose that p57+ cells are a unique population contributing to injury induced plasticity, while on the other hand, the data seemed to overlap these cells substantially with Tuft and EE cells, both of which have been previously shown to contribute to injury-induced plasticity. It was not entirely clear what was new, in terms of plasticity, other than a new marker (p57) for these cell populations.
4. Fig. 6 showed interesting gastric metaplastic changes upon severe injury, however the data appear isolated from the theme of p57+ cells?

Specific comments:

- Fig. 1f and 1i are inconsistent.
- Fig. 2h, what about other genes such as Bmi1, Lrig1, HopX, Clu, and Lyz1, etc.
- Fig. 2: The author concluded by stating "Collectively, these results suggested that, although p57-expressing cell populations share the quiescence gene program with other tissue stem cells, they behave as EE/tuft lineage cells rather than as undifferentiated, "professional" quiescent ISCs." Without testing the function of these p57 cells, it was unclear how the authors reached the conclusion that these cells "behave" as EE/tuft lineage but not "professional" quiescent ISCs?
- Line 186: "Given that the frequency of Venus+ cells in p57- DTR-Venus mice was ~1 cell per 20 crypts, we estimate that ~20% of total p57+ cells in intestinal crypts are labeled in this lineage tracing system." What was the frequency of endogenous p57 cells? Why not use that (p57 staining shown in Fig. 2g) as a reference?
- Fig. 3i, was the ribbon also observed in duodenum? What was the frequency compared to jejunum and ileum?

- Fig. 3l-n, can this Wnt-activated plasticity be demonstrated in vivo?
- Authors stated: line 225: "these results implicated the activation of Wnt signaling in the reversion of p57+ cells to functional ISCs."
Line 230: "quiescent p57+ cells dedifferentiate from their EE/tuft cell precursor state and replenish Lgr5high CBCs lost as a result of crypt injury."
Are the authors suggesting that injury activated Wnt signaling, which activated plasticity? If yes, then how did injury activate Wnt signaling?
- Line 233: scRNA-seq analysis with the CEL-seq2 platform of p57-Venus+ and p57-Venus- cells isolated from crypts of p57-DTR-Venus mice.
Were these scRNA from ileal or duodenal crypts? The authors should include these information in main text.
- Fig. 4b, was the data also suggesting that a fraction (on the left) of Paneth cells expressed p57-Venus in 5-FU day 3 crypts?
- Fig. 4h, the events of Sox2 cells seemed very rare, and not consistent with the statement on line 281.
- Fig. 5b, the cluster 2 and 19 "revival p57+ cells" contained a noticeable portion of p57-venus-negative cells.
- Fig. 5c, the number (or relative proportion) of postinjury p57-venus-positive cells is different from Fig. 5a and Fig. 4b.
- Fig. 6, are these gastric metaplastic changes dependent on p57 cells?
- Fig. 7e-h, are these images of colon?

Reviewer #2 (Remarks to the Author):

The manuscript by Higa et al, entitled "Spatiotemporal reprogramming of differentiated cells underlies regeneration and neoplasia in the intestinal epithelium", identifies p57 as a functional marker for quiescent cells that can serve as facultative stem cells. These cells are progenitors to secretory cells (Prox1/Bmi1 cells in Yan et al.) at homeostasis and gets activated to become revival stem cells during damage (Wrana lab's paper). This is a well-done manuscript and its claims are well supported at every step by experiments. These are the few comments from us:

1. Figure 2 (top). The authors spent quite a bit of time and space on re-analyzing Yan et al. scRNA-seq data. This re-analysis did not provide new data or new insights since Yan et al already showed that Bmi and Prox1 marked quiescent stem cells. Aside from p57 expression in those cells, additional re-analysis of old data should be in supplement.
2. The authors demonstrated that p57 cells activate/regain stem cell properties upon damage (Irradiation/Lgr5 ablation/ 5-FU). They authors implicate a metaplasia program for reactivation of stem cell capacity in the experiments following. However, they only did follow-up studies on 5-FU and derivative conditions (FP). It is unclear if metaplasia is an universal mechanism to reactivate stem cells.
3. What happens if p57+ cells are ablated in different conditions, especially in conditions of damage? Seems like this question is readily answerable with the p57-DTR allele reported in this paper.
4. It would be appropriate to include appropriate controls for LGR5+ cell organoid formation (such as with ENRWC) – Figure 3 m,n.

5. There is a concern about the scRNA-seq data generated from this study. The data clustered by condition (Fig. 4) with no intermixing. It is unclear if the differences are generated by technical effects or biological effects. While the author stated it is n=2 per condition, it is unclear whether it is n=2 pooled together and ran as n=1 sample, or if it is n=2 independent experiments (separate runs). It is important for multiple biological replicates to be performed independently to make sure the findings are reproducible and not resulting from technical effects.

6. For RACEID2, please overlay on the same coordinates in figure 4. Unclear which cells belong to which condition. It's unclear whether they are different subsets of cells, but we could infer that they are different cell subsets since they seem to use two different filtering methods, one for t-SNE visualization and one for StemID. These filtering methods should be made consistent, if the comparison between the two tSNEs is going to be made. For example, there was a mitochondrial filter for Seurat-based tSNE but not for the StemID tSNE.

7. The authors imply of Crohn's-like mechanism in metaplasia, which is shown to be epithelial cell-intrinsic with some overlaid signatures and histology. However, damage and inflammation are usually more complex than cell-intrinsic patterns and there should be at least some basic immune cell analyses to accompany the figures used in Figure 6.

8. CDX2 and CDX1 have some shared interactions, with respect to cellular plasticity. CDX2 should also be highlighted or mentioned, as it has some implications in neoplasias.

Analysis comments:

9. Why, in 5c do there seem to be cell types that span multiple, seemingly disparate unsupervised clusters? This is the case for some enterocytes, paneth, CBC. Is there some signature difference between the different parts of these split-up clusters or potentially an artifact of Harmony batch correction? The authors should include some uncorrected scRNA-seq figures to compare.

10. Some of the gene signatures examined have overlapping genes, so please consider the effect that these overlaps will have. The authors should ensure that these overlaps do not lead to misleading statistical test results.

11. Figure 5a and b, entropy seems to be approximately between 1 and .5, why were these values not value scaled (i.e. start from 0), what is the nature of its distribution, and are these distributions significantly different by cluster or p57-Venus condition? I didn't see any statistical tests for these, just qualitative bar plots. Something like a violin plot per cluster or per sample type would be more interpretable.

12. Figure 5a, the linkage edges are very difficult to interpret, why are some linked to ones across the tSNE? Is this due to some nonlinear reduction effect from tSNE and does this unusual linkage patterning maintain in PCA or UMAP space?

13. Should show more than just Ki67 to the scRNA-seq analysis, consider published lists of different phase signatures, Seurat has a built-in cell cycle scoring that would strengthen the cell cycle claims for scRNA-seq figures.

Minor

- For Figure 1m, this could be better complemented by a cell cycle analysis instead of just ki67 staining.
- Need to make Fig 2bd and 4 t-SNEs consistent in terms of dot scaling size.
- Figure 5, could use a heatmap of scaled entropy scores per single cell. Difficult to interpret the 'lineage tree'.
- The authors should comment on the methodologies used to quantify the crypt numbers, like in Figure 6. Were they randomly selected fields of view or evaluated blind? Is 1+ cell enough to score etc.?
- Fig 6a is not referenced anywhere, please ensure that all figures and subfigures are properly referenced.

Reviewer #3 (Remarks to the Author):

In this study Higa and colleagues investigate the ability of differentiated intestinal cells to reprogramme upon injury. Using a number of in vivo models, they define a population of cells marked by p57 as enteroendocrine/tuft cell precursors that are able to dedifferentiate and acquire stem cell properties upon damage. They further utilise single cell RNAseq to characterise the lineage changes these cells undergo and their reprogramming into a fetal/regenerative like state. They also observe similar characteristics in intestinal adenomas. Overall, this is an interesting study, which uses a number of elegant models to make some interesting observations on the cell type changes that occur during regeneration. Although some of the findings have previously been observed (fetal reprogramming etc) the precision with which the authors map the process certainly adds to our understanding of it and is a useful addition to the field. I have a number of comments that the authors should address prior to publication:

- 1) The key take home message of the paper is that p57+ cells reprogramme and repopulate the intestine following injury. However, it is not clear to me how important p57+ cells are for regeneration as the authors do not directly test the consequences of depleting them in their models. This is important as in Figure 6 the authors suggest additional cell types also gain regenerative marker expression upon injury suggesting that many/all cell types in the intestine may have regenerative potential. It is possible that p57+ cells are simply a rare population of cells with no special properties different from other differentiated cell populations. The authors should investigate the functional consequences of depleting p57+ cells on the ability of the intestine to regenerate following injury. For example, they could utilise their p57-DTR model to deplete this cell population (alongside 5FU treatment or crossed to the Lgr5-DTR model).
- 2) The single cell data (Fig 4c, d + e) suggests the majority of regenerative cells are derived from p57 Venus+ cells. Why then is the regenerative potential of p57+ cells so low (Figure 3J - ~15 events per SI)?
- 3) Fig 5 is an interesting analysis but would benefit from functional data. The authors could sort Venus+ cells from 5FU treated mice and determine whether they can form organoids in vitro.
- 4) Figure 1j-n characterises the effects of p57 ko on the dynamics of Bmi1+ cells suggesting it maintains their quiescence. Again, functional characterisation of these cells would be helpful – do they gain stem cell properties (organoid forming capacity) upon escape from quiescence?

Response to Reviewer #1

We thank the reviewer for the careful appraisal of our manuscript and for the statements that “The interpretation of most of the data was appropriate” and that “The notion that gastric metaplastic change may occur not only during injury-induced plasticity but in intestinal tumorigenesis is interesting.” We also thank the reviewer for several suggestions that we feel have helped us to improve our manuscript.

Our specific responses to the points raised are as follows:

Major concerns:

1-1. The authors used p57-CreER mice to demonstrate injury induced lineage tracing in Fig. 3, but used p57-Venus mice for scRNA-seq analysis of injury induced plasticity. As the Venus is controlled under the endogenous p57 promoter (although it can be self-cleaved), and injury turned endogenous p57 gene off (stated by authors on line 242), the scRNA results are expected to be somewhat confounded by such injury-induced alteration of p57 promoter thereby Venus+ cells.

[Response] We apologize for the inadequate description that might have led to some misunderstanding. As mentioned in the original manuscript (pages 8–9, lines 242–253), although p57 gene expression declined below the threshold for detection by scRNA-seq analysis after injury (**new Fig. 4b and Supplementary Fig. 4f**), the abundance of the Venus protein was maintained even in the postinjury p57⁺ cells (**Supplementary Fig. 4a**), likely as a result of its marked stability in mammalian cells (ref. 32). Consistent with this notion, RT-qPCR analysis showed that p57 mRNA was partially down-regulated but still selectively present in the Venus⁺ fraction after injury (**new Supplementary Fig. 4h**). We therefore concluded that the scRNA-seq data set faithfully recapitulates the kinetics of p57⁺ cells undergoing regeneration. To avoid any misunderstanding, we have now modified the corresponding description in the revised manuscript (page 10, lines 295–308).

Higa et al., Figure 4

Higa et al.,
Supplementary Figure 4

As the reviewer points out, it might have been reasonable to use p57-CreER^{T2}/CAG-LSL-tdTomato mice for the scRNA-seq experiments if we had wanted to obtain the data under conditions identical to those for the lineage tracing experiments. However, these mice require tamoxifen injection to label the cells of interest, which has been shown to damage epithelial cells in various tissues (Leushacke *et al.*, *Nat. Cell Biol.* 19: 774–86, 2017; Yang *et al.*, *Pharmacol. Ther.* 139: 392–404, 2013). Given that scRNA-seq analysis is readily influenced by technical variability, we opted to exclude unwanted and unexpected technical artifacts as much as possible in order to perform a precise evaluation of postinjury transcriptomic changes. In addition, the p57⁺ cells labeled in p57-CreER^{T2}/CAG-LSL-tdTomato mice are much rarer (~0.1%) than are p57-Venus⁺ cells (~0.5%) (**new Supplementary Fig. 2c**), as we now describe in the revised manuscript (page 7, lines 189–195). This discrepancy is likely attributable to the limited efficiency of CreER^{T2} activation by tamoxifen, a problem often found in similar lineage tracing experiments (Barriga *et al.*, *Cell Stem Cell* 20: 801–816, 2017). Single-cell sorting of such a small proportion of cells would thus be time-consuming and result in evaporation of the lysis solution (only 1.2 μ l) initially applied to the wells of the 96-well plate, which would lead to loss of data quality. Given these general and study-specific technical issues, we concluded that the use of p57-CreER^{T2}/CAG-LSL-tdTomato mice was inappropriate or impractical for scRNA-seq analysis, and decided to use p57-Venus mice instead.

Higa et al., Supplementary Figure 2

1-2. It was not shown whether p57-Venus mice also formed ribbons after injury, as p57-CreER mice.

[Response] p57-Venus mice allow visualization of p57⁺ cells but not labeling of their progeny, and it is impossible in principle to check whether they form ribbons in vivo. As an alternative approach to examine whether p57-Venus⁺ cells acquire stemness after injury, we cultured FACS-sorted normal or postinjury p57-Venus⁺ crypt cells in vitro. We found that the postinjury p57-Venus⁺ cells formed intestinal organoids, whereas control p57-Venus⁺ cells almost completely lacked this ability (**new Fig. 5c, d**). These results suggest that p57-Venus⁺ crypt cells indeed acquire stemness after injury, as is the case for p57-CreER^{T2}-tdTomato⁺ cells, validating the use of p57-Venus mice for our scRNA-seq experiments. We have now clarified these points in the revised manuscript (page 12, lines 374–379).

Higa et al., Figure 5

2. Fig. 4b was a little confusing as well as surprising. These scRNA-seq were from crypt cells. The Tuft and EE cells appeared to be great over-represented in tSNE plots of normal crypts than expected. In contrast, the number of Paneth cells, which are major crypt cells, seemed surprisingly under-represented in the tSNE of normal crypts. By comparing the two panels in 4b, it almost indicated that virtually all Paneth cells were only present in 3-day 5-FU treated mice, and they were absent from normal crypts. In addition, authors stated that p57-Venus highlights 1 cell per 20 crypts (line 186). If this was the case, then the proportion of p57-venus-positive vs p57-venus-negative cells represented in these tSNE plots in normal crypts was problematic. These discrepancies raise questions about how the scRNA data were analyzed, and whether the analysis removed substantial numbers of cells from the tSNE plots.

[Response] We again apologize for the insufficient explanation of the t-SNE plots, which may have caused some confusion. We would like to explain these issues in the following two parts:

(1) With regard to the proportion of p57⁺ and p57⁻ cells

In the scRNA-seq data of the original manuscript, we sorted almost equal numbers of p57-Venus⁺ and p57-Venus⁻ cells and subjected them to library preparation, given that we initially intended to investigate the postinjury transcriptomic changes of p57⁺ cells including their heterogeneity. The resultant plot therefore did not reflect the actual proportion of each cell type in intestinal crypts in vivo. Similar approaches have been widely adopted in previous scRNA-seq studies of tissue stem cells including ISCs, especially in order to

investigate rare cell populations of interest (ref. 14; Pellin *et al.*, *Nat. Commun.* 10: 2395, 2019; Dong *et al.*, *Nat. Cell Biol.* 22: 630–9, 2020), and we do not think that our initial experimental design was inappropriate. Nevertheless, we decided that such representation of the data might lead to the misunderstanding that p57⁺ cells constitute the major cell population of intestinal crypts, as the reviewer points out. We therefore newly obtained scRNA-seq data from additional mouse samples (replicate 2), with more p57-Venus⁻ cells than p57-Venus⁺ cells being included for the analysis in this replicate. In the revised manuscript, we analyzed these data together with our original data (replicate 1) in an integrated manner (**new Fig. 4 and Supplementary Fig. 4b**), and we confirmed that the key findings of our study (spatiotemporal reprogramming and postinjury stemness acquisition) were apparent in both replicates. We believe that this approach mitigates the extreme enrichment of p57⁺ cells as well as ensures the reproducibility of our findings. Although the proportion of p57⁺ cells on the *t*-SNE plot remains higher than that in actual intestinal crypts, we believe that it is impractical for the scRNA-seq analysis to be performed without enrichment of p57-Venus⁺ cells, given the small size of this cell population. We have now clarified the methodology and results of our new scRNA-seq analysis in both the Results and Methods sections of the revised manuscript (pages 9–11, lines 267–354; pages 21–22, lines 707–721).

(2) With regard to the lack or enrichment of Paneth cells

The reviewer also points out the lack or enrichment of Paneth cells in normal or postinjury crypt cells, respectively. Postinjury enrichment of Paneth-like cells has also been described in a previous scRNA-seq study based on Lgr5^{high} cell ablation (ref. 16), indicating that this might be a common feature of the injured intestinal epithelium. The absence of Paneth cells in the normal crypt cell population of our original data (replicate 1) is likely attributable to the low number of the cells sequenced. In our new scRNA-seq data in the revised manuscript, the Paneth cell clusters contained not only postinjury p57-Venus⁻ cells but also normal p57-Venus⁻ and p57-Venus⁺ cells (**new Fig. 4b**). We have now addressed these points in the revised manuscript (page 9–10, lines 280–286).

Higa et al. Figure 4

Higa et al., Supplementary Figure 4

3. Conceptually, on one hand, the authors tried to propose that p57⁺ cells are a unique population contributing to injury induced plasticity, while on the other hand, the data seemed to overlap these cells substantially with Tuft and EE cells, both of which have been previously shown to contribute to injury-induced plasticity. It was not entirely clear what was new, in terms of plasticity, other than a new marker (p57) for these cell populations.

4. Fig. 6 showed interesting gastric metaplastic changes upon severe injury, however the data appear isolated from the theme of p57⁺ cells?

[Response] These two comments are closely related, and we would therefore like to address them together. As pointed out by the reviewer in comment 3, the notion of EE/tuft cell plasticity itself is not novel. As discussed in the original and revised (page 15, lines 477–486) versions of the manuscript, we believe that one of the most powerful features of p57 as a tracing marker is its prominent specificity for the quiescent cells around the +4 position. As addressed in earlier studies, the identity and stem cell activity of +4 ISCs have been controversial, given the lack of specific tracing markers for these cells (Barker, *Nat. Rev. Mol. Cell Biol.* 15: 19–33, 2013; refs. 11, 12). At least from one view, the results of our p57 tracing experiments support the notion that +4 ISCs are EE/tuft cell precursors and that they serve as injury-induced (not “professional”) ISCs. Given the long-standing debate regarding the identity of +4 ISCs, our data provide important insight into ISC systems.

We believe that a key finding of our study is the detection of spatiotemporal reprogramming, an event revealed initially by the scRNA-seq analysis of the reversion process for p57⁺ cells. In particular, this finding links the postinjury reversion of crypt cells to the metaplasia-like changes, which were subsequently shown to be recapitulated also in adenoma. We consider that our scRNA-seq approach focusing on a specific cell type (p57⁺ cells) with a greater sequencing depth ($\sim 10^5$ reads per cell) (page 22, lines 730–733) allowed us to uncover this phenomenon, given that it has not been detected in any previous high-throughput scRNA-seq studies (refs. 18, 20). Subsequent population-level analyses showed that such metaplasia-like changes are universal events occurring also in the other cell types (**new Fig. 6**). Given that several crypt cell types have been shown to dedifferentiate after injury, it is likely that such a postinjury mechanism commonly underlies the reversion process of different crypt cell types. With regard to comment 4, we believe that such universality of this phenomenon, rather than its specificity for p57⁺ cells, is an important observation.

The processes of postinjury reversion and cancer initiation in intestinal crypts remain poorly

understood. Exploration of the molecular mechanisms underlying spatiotemporal reprogramming should lead to a detailed understanding of such reversion and oncogenic processes. In this regard, we believe that our present findings provide a novel and meaningful basis for future studies. We have now modified the text to underscore these points in both the Results and Discussion sections of the revised manuscript (page 14, lines 430–432; page 15, lines 497–499).

Higa et al. Figure 6

Specific comments:

- Fig. 1f and 1i are inconsistent.

[Response] We interpreted this comment as follows: *Figure 1f suggests that Bmi1-expressing cells and p57-expressing cells are almost completely overlapping cell types. In Figure 1i, however, despite the fact that p57 mRNA is present only in Bmi1^{high} cells (being almost completely absent in Lgr5^{high} cells), the abundance of Bmi1 mRNA is almost equal in both Bmi1^{high} cells and Lgr5^{high} cells. The two figure panels thus appear inconsistent.*

We believe that this apparent inconsistency is attributable to the discrepancy between Bmi1 mRNA (**Fig. 1i**) and its GFP reporter expression (**Fig. 1f**) in intestinal crypts. Although Bmi1 was originally described as a marker for quiescent +4 ISC (refs. 3, 6, 9), many previous studies have shown that Bmi1 mRNA is present in all crypt cells including Lgr5^{high} CBCs (refs. 11, 12). In contrast, expression of the corresponding GFP reporter is highly restricted to quiescent cells around the +4 position. Indeed, our immunofluorescence analysis of Bmi1-GFP mouse intestine revealed rare Ki67⁻ quiescent cells with a relatively high GFP signal around the crypt +4 position (**Fig. R1, for reviewing purposes only**). Previous studies such as reference 3 also used Bmi1-GFP mice to identify +4 ISCs, which are likely identical to the cell population we observed. Consistent with the notion that Bmi1-GFP^{high} cells are a distinct cellular entity, these cells indeed frequently overlap with p57⁺ cells in immunofluorescence analysis (**Fig. 1e, f**). Expression of p57 was also specific to FACS-sorted Bmi1-GFP^{high} cells at the mRNA level (**Fig. 1h, i**). Collectively, these findings indicate that the expression patterns of Bmi1-GFP and Bmi1 mRNA are not identical, with the former being relatively specific to quiescent cells around the +4 position and the latter being promiscuously expressed also in Lgr5^{high} CBCs (**Fig. 1i**).

Higa et al., Figure 1

a**Higa et al. Figure R1**

Figure R1 | Three representative immunofluorescence images of GFP and Ki67 in crypts of the small intestine of Bmi1-GFP mice. Arrowheads indicate Ki67⁻ GFP-positive cells.

- Fig. 2h, what about other genes such as *Bmi1*, *Lrig1*, *HopX*, *Clu*, and *Lyz1*, etc.

[Response] RT-qPCR analysis revealed that the expression of classical +4 markers (*Bmi1*, *Lrig1*, and *Hopx*) did not show significant enrichment in p57-Venus⁺ cells compared with whole-crypt cells (**new Supplementary Fig. 2b**), likely as a result of their substantial expression in other crypt cell types including *Lgr5*^{high} CBCs (ref. 12). The abundance of *Clu* mRNA was significantly higher in p57-Venus⁺ cells than in whole-crypt cells (**new Supplementary Fig. 2b**), indicating that the normal p57⁺ cells overlap at least in part with the preexisting *Clu*⁺ cells in normal intestinal crypts (ref. 20). We also now show mRNA abundance for the Paneth cell markers *Defa5* and *Mmp7* (**new Supplementary Fig. 2b**), instead of *Lyz1*. As mentioned above, although normal p57-Venus⁺ clusters contained a few Paneth-like cells in our scRNA-seq data set, we concluded overall that p57-Venus⁺ cells are precursors of EE/tuft cells but not of Paneth cells, given that repeated RT-qPCR analysis did not reveal significant enrichment of these Paneth cell markers in p57-Venus⁺ cells. We have now clarified these points in the revised manuscript (page 6, lines 174–177; page 10, lines 289–295).

b**Higa et al., Supplementary Figure 2**

- Fig. 2: The author concluded by stating “Collectively, these results suggested that, although p57-expressing cell populations share the quiescence gene program with other

tissue stem cells, they behave as EE/tuft lineage cells rather than as undifferentiated, “professional” quiescent ISCs.” Without testing the function of these p57 cells, it was unclear how the authors reached the conclusion that these cells “behave” as EE/tuft lineage but not “professional” quiescent ISCs?

[Response] We agree with the reviewer’s comment and apologize for our inappropriate description of the conclusion drawn from the scRNA-seq data analysis. As the reviewer points out, it is an overstatement to conclude that p57⁺ cells “behave” as EE/tuft lineage cells without testing their stem cell activity. We believe that we are able to state only that these cells “can be classified as” EE/tuft lineage cells rather than as undifferentiated ISCs on the basis of the single-cell transcriptomics analysis. We have now corrected this sentence in the revised manuscript (pages 6–7, lines 177–180).

- Line 186: “Given that the frequency of Venus⁺ cells in p57- DTR-Venus mice was ~1 cell per 20 crypts, we estimate that ~20% of total p57⁺ cells in intestinal crypts are labeled in this lineage tracing system.” What was the frequency of endogenous p57 cells? Why not use that (p57 staining shown in Fig. 2g) as a reference?

[Response] In response to this comment, we performed immunostaining and quantification of “endogenous” p57⁺ cells, and found that their frequency was almost equal to that of p57-Venus⁺ cells (**new Supplementary Fig. 2c**). We have now described these results in the revised manuscript (page 7, lines 189–195).

Higa et al., Supplementary Figure 2

- Fig. 3i, was the ribbon also observed in duodenum? What was the frequency compared to jejunum and ileum?

[Response] We observed tracing ribbons also in the duodenum (**new Supplementary Fig. 2e**), and their frequency did not differ significantly among the duodenum, jejunum, and ileum (data not shown). We have now clarified this point in the revised manuscript (pages 7–8, lines 213–217).

Higa et al., Supplementary Figure 2

- Fig. 3l-n, can this Wnt-activated plasticity be demonstrated in vivo?

[Response] Intraperitoneal injection of mice with CHIR99021, a GSK3 inhibitor that activates the Wnt signaling pathway, has been shown to up-regulate Wnt signaling in intestinal crypts without any injury context (ref. 29). To investigate whether Wnt activation is sufficient to induce the reversion of differentiated cells in vivo, we injected p57-CreER^{T2}/CAG-LSL-tdTomato mice intraperitoneally with tamoxifen (50 mg/kg) and CHIR99021 (1 mg/kg). After 14 days, we detected tracing events in these mice, although their frequency was lower than that in those subjected to intestinal injury (**new Fig. 3j**). These results suggest that CHIR99021-mediated Wnt activation is sufficient to induce dedifferentiation of p57⁺ cells, albeit to a limited extent compared with injury contexts. The difference in the frequency of lineage tracing events between CHIR99021 treatment and injury contexts might be explained by the difference in long-term availability of the ISC niche space. Given that Lgr5^{high} CBCs are eliminated after injury, p57⁺ cells may be able to reside in the niche space and to receive niche factors that maintain stemness (refs. 15, 16) for the long term. On the other hand, given that CHIR99021-mediated Wnt activation does not damage Lgr5^{high} CBCs, it is likely that the induction of stemness by CHIR99021 in p57⁺ cells is transient, and that the cells have less of a chance to function as long-term ISCs compared with the situation for injury-induced dedifferentiation. We have now addressed these issues in the revised manuscript (page 8, lines 235–243).

Higa et al., Figure 3

- Authors stated: line 225: “these results implicated the activation of Wnt signaling in the reversion of p57⁺ cells to functional ISCs.”

Line 230: “quiescent p57⁺ cells dedifferentiate from their EE/tuft cell precursor state and replenish Lgr5^{high} CBCs lost as a result of crypt injury.”

Are the authors suggesting that injury activated Wnt signaling, which activated plasticity? If yes, then how did injury activate Wnt signaling?

[Response] To examine whether Wnt signaling is activated after 5-FU–induced injury, we injected WT mice intraperitoneally with vehicle or 5-FU (150 mg/kg). RT-qPCR analysis on day 3 showed that representative Wnt reporter genes in the intestine (*Cd44*, *Cd133*, and *Axin2*) were significantly up-regulated in 5-FU–treated whole-crypt cells compared with control cells (**new Supplementary Fig. 3a**), suggesting that Wnt signaling is indeed activated after 5-FU treatment. Representative Wnt-regulated CBC-specific genes such as *Lgr5* and *Ascl2* were also up-regulated in whole-crypt cells after 5-FU treatment (**new Supplementary Fig. 3a**). Given that 5-FU almost completely eliminates Lgr5^{high} CBCs at this time point (**new Supplementary Fig. 2d**), these results are indicative of the occurrence of a dedifferentiation event and activation of an ISC gene program in the lineage-committed cell populations after injury. We believe that these results are consistent with those of previous studies showing that intestinal injury such as that induced by irradiation (Suh *et al.*, *Cell Rep.* 21: 2571–84, 2017) or ablation of Lgr5^{high} CBCs (Murata *et al.*, *Cell Stem Cell* 5: 377–90, 2020) activates Wnt signaling in crypts. We have now addressed this issue in the revised manuscript (page 8, lines 223–225). The data showing the Wnt-induced dedifferentiation of p57⁺ cells in **Figure 3l–n** of the original manuscript were moved to **new Supplementary Figure 3b–d** in the revised manuscript.

With regard to the mechanism underlying Wnt activation after intestinal injury, Clevers’ group has proposed the “vacant niche theory” (ref. 16), which states that differentiated crypt cells are able to enter the ISC niche at the crypt base after the loss of CBCs and become exposed to niche factors such as Wnt that activate the dedifferentiation program. We anticipate that this well-accepted theory is also applicable to the dedifferentiation of p57⁺ cells. On the other hand, however, recent studies have implicated other signaling pathways such as the Hippo (refs. 19, 20) and interferon (ref. 18) pathways in postinjury dedifferentiation events, raising the possibility that injury-induced reversion in vivo is not sufficiently explained by niche signaling factors alone. Given that the mechanism underlying activation of these pathways and the extent to which they are responsible for the induction of reversion are unknown and beyond the scope of the present study, we would like to focus on such questions in the future.

Higa et al., Supplementary Figure 3

Higa et al., Supplementary Figure 2

- Line 233: scRNA-seq analysis with the CEL-seq2 platform of p57-Venus⁺ and p57-Venus⁻ cells isolated from crypts of p57-DTR-Venus mice. Were these scRNA from ileal or duodenal crypts? The authors should include these information in main text.

[Response] We apologize for the insufficient description regarding the origin of crypt cells subjected to scRNA-seq analysis. We isolated whole-crypt cells from all regions of the small intestine (duodenum, jejunum, and ileum) for the scRNA-seq analysis. Although it might have been better for this analysis to be performed separately with the different regions of the intestinal epithelium, we judged that this was impractical given that the low frequency of p57⁺ cells would render their isolation in sufficient numbers problematic. We have now clarified this point in both the Results and Methods sections of the revised manuscript (page 9, lines 271–275; page 21, lines 707–709).

- Fig. 4b, was the data also suggesting that a fraction (on the left) of Paneth cells expressed p57-Venus in 5-FU day 3 crypts?

[Response] As suggested by the reviewer, a small fraction of normal and postinjury p57⁺ cells included Paneth-like cells in the scRNA-seq analysis of both replicate 1 and replicate 2 (**new Fig. 4b and Supplementary Fig. 4b**). Together with our data showing that p57⁻-expressing cells are quiescent, located around the +4 position, and enriched for the expression of LRC signature genes (**new Fig. 2e**), the existence of p57-Venus⁺ Paneth-like cells revealed by the scRNA-seq data appears to be consistent with the previous finding that LRCs serve as both EE and Paneth cell precursors (ref. 13). Nevertheless, given that enrichment of Paneth cell genes such as *Defa5* and *Mmp7* was not apparent in p57-Venus⁺ cells in repeated population-level RT-qPCR analysis (**new Supplementary Fig. 2b**), we overall consider that most p57-Venus⁺ cells are EE/tuft cell precursors in our study. We have now clarified this point in the revised manuscript (page 10, lines 289–295). Given that our study mainly focuses on postinjury reversion and spatiotemporal reprogramming of p57⁺ or other crypt cells, we plan to investigate the detailed differentiation path or heterogeneity of the p57⁺ cell population in future studies.

Higa et al., Figure 4

Higa et al., Supplementary Figure 4

Higa et al., Figure 2

- Fig. 4h, the events of Sox2 cells seemed very rare, and not consistent with the statement on line 281.

[Response] We agree with the reviewer on this point, and we have now omitted the data for cells expressing Sox2 in the revised manuscript.

- Fig. 5b, the cluster 2 and 19 “revival p57⁺ cells” contained a noticeable portion of p57-venus-negative cells.

[Response] Given that another reviewer requested that we calculate the stemness score for each single cell, we applied CytoTRACE software instead of StemID analysis. Our revised manuscript does not contain data like those of Figure 5b in the original version. The reviewer’s comment is thus no longer valid. Of note, we decided not to use the term “revival p57⁺ cells” in the revised manuscript, but instead adopted the terms “Regenerating 1” for clusters consisting mainly of postinjury p57-Venus⁺ cells and “Regenerating 2” for those comprising mostly postinjury p57-Venus⁻ cells.

- Fig. 5c, the number (or relative proportion) of postinjury p57-venus-positive cells is different from Fig. 5a and Fig. 4b.

[Response] The original Figure 5c showed an integrative analysis of our scRNA-seq data together with the data from reference 14, and the relative ratio of postinjury p57-Venus⁺ cells to the other cells is in principle different from that in the original Figures 4b and 5a. The absolute number of p57-Venus⁺ cells did not differ between the original Figure 5c and Figures 4b and 5a, given that we did not change the filtering threshold. In the revised manuscript, however, we omitted such an integrative analysis, given that we have now added new data for a substantial number of p57-Venus⁻ cells for comparison with p57-Venus⁺ cells. The reviewer’s comment is thus no longer relevant.

- Fig. 6, are these gastric metaplastic changes dependent on p57 cells?

[Response] We initially sought to investigate the requirement for p57⁺ cells in postinjury intestinal regeneration through in vivo ablation of these cells. As we mention in the text

(page 8, lines 244–249), however, we were unable to examine the long-term effects of p57⁺ cell ablation in vivo because of the mortality of p57-DTR-Venus mice after DT injection. Given the high expression level of p57 in adrenal glands (Nagahama *et al.*, *Anat. Embryol.* 203: 77–87, 2001), we speculate that acute adrenal failure might be responsible for the mortality. As a short-term analysis, we injected p57-DTR-Venus mice with DT 3 days after 5-FU treatment and assessed crypt viability 24 h later (**Fig. R2a, b, for reviewing purposes only**) by immunostaining for Ki67, as previously described (Metcalf *et al.*, *Cell Stem Cell* 14: 149–59, 2014). We found that the postinjury regenerative response apparent in control mice was markedly impaired in mice deficient in p57⁺ cells, as indicated by a significant decrease in the number of viable crypts (**Fig. R2c–e, for reviewing purposes only**). Although these results are suggestive of an indispensable role for crypt p57⁺ cells in postinjury epithelial regeneration, we cannot exclude the possibility that such a phenotype was the consequence of intestine- or epithelium-extrinsic effects, given that this experimental system results in the ablation of p57⁺ cells in all tissues and the mice were morbid. Taking these points into consideration, we decided not to include these data in our revised manuscript.

For the same reason as mentioned above, we consider that it is currently impractical for us to examine whether the metaplasia-like changes are dependent on p57⁺ cells. Nevertheless, we anticipate from lines of indirect evidence that postinjury spatiotemporal reprogramming, including the metaplasia-like changes, is not dependent solely on p57⁺ cells, given that these changes were also apparent in other cell types, as shown in Figure 6. As we mentioned in the response to major concerns 3 and 4, we consider that such universality, rather than specificity for p57⁺ cells, makes this finding more important, given that postinjury dedifferentiation is likely a common property of all crypt cell types. We have now emphasized these points in the revised manuscript (page 14, lines 430–432; page 15, lines 497–499).

Higa *et al.* Figure R2

Figure R2 | **a** FACS plots for isolation of Venus⁺ cells from intestinal crypts either of p57-DTR-Venus mice injected with PBS or DT (50 µg/kg) 24 h before analysis or of WT mice as a negative control. **b** Dosing regimen for p57-DTR-Venus mice subjected to ablation of p57⁺ cells after the induction of crypt injury by injection of 5-FU (150 mg/kg). **c, d** Representative immunofluorescence staining of Ki67 in the intestinal epithelium at 4 days after injury induction without (**c**) or with (**d**) subsequent ablation of p57⁺ cells in p57-DTR-Venus mice. The active regenerative response apparent in **c** is impaired in **d**. Scale bars, 50 µm. **e** Box plots showing the number of viable crypts per field (1-mm length of the intestine) at 4 days after the induction of injury with or without subsequent p57⁺ cell ablation in p57-DTR-Venus mice as in **c** and **d**. Regenerating crypts were defined as cryptlike structures containing ≥5 adjacent Ki67⁺ cells, as previously described (Metcalf *et al.*, 2014). About 20 fields of view were counted per mouse (*n* = 3 mice). The box plots represent the maximum, upper quartile, lower quartile, and minimum values, with the horizontal bars corresponding to the median. *****P* < 0.001 (Student's *t* test).

Although ablation of p57⁺ cells *in vivo* is technically difficult, we investigated the effect of p57⁺ cell ablation with the use of p57-Venus intestinal organoids *in vitro*, given that this approach allowed us to exclude the possible involvement of intestine- or epithelium-extrinsic effects. We found that combined treatment with 5-FU and DT markedly impaired organoid formation compared with that with 5-FU or DT alone (**new Fig. 3l–n**). These results thus suggested that p57⁺ cells make a limited contribution to homeostatic epithelial maintenance but are required for postinjury regeneration. We have now addressed these issues in the revised manuscript (pages 8–9, lines 249–265).

Higa *et al.*, Figure 3

- Fig. 7e-h, are these images of colon?

[Response] All images in Figure 7 represent adenoma arising in the small intestine. We have now clarified this point in the revised manuscript (page 14, lines 441–442).

Response to Reviewer #2

We thank the reviewer for the careful evaluation of our manuscript and for the statement that “This is a well-done manuscript and its claims are well supported at every step by experiments.” We also thank the reviewer for several suggestions that we feel have helped us to improve our manuscript.

Our specific responses to the points raised are as follows:

1. Figure 2 (top). The authors spent quite a bit of time and space on re-analyzing Yan et al. scRNA-seq data. This re-analysis did not provide new data or new insights since Yan et al already showed that Bmi and Prox1 marked quiescent stem cells. Aside from p57 expression in those cells, additional re-analysis of old data should be in supplement.

[Response] Although we respect the reviewer’s comment, we believe that the data in Figure 2 are vital to set the logical framework of our study. Given that p57 has not previously been mentioned as a marker for homeostatic or facultative ISCs, we believe that the analysis in Figure 2 is essential in order for us to address p57⁺ cell identity in relation to previously well-characterized cell populations (such as LRCs or Prox1⁺ or Bmi1^{high} cells) with the use of accepted scRNA-seq data in the public domain. Taking these points into consideration, we prefer to present these data as a main figure in our revised manuscript.

2. The authors demonstrated that p57 cells activate/regain stem cell properties upon damage (Irradiation/Lgr5 ablation/ 5-FU). They authors implicate a metaplasia program for reactivation of stem cell capacity in the experiments following. However, they only did follow-up studies on 5-FU and derivative conditions (FP). It Is unclear if metaplasia is an universal mechanism to reactivate stem cells.

[Response] In response to this comment, we examined whether spatiotemporal reprogramming, in particular metaplasia-like changes, is a universal mechanism for reactivation of stem cells by subjecting WT mice to whole-body irradiation. Immunofluorescence analysis of these mice revealed that Pgc and Tff2—markers for gastric chief and mucous cells, respectively—were expressed in the jejunum and ileum of these mice (**new Supplementary Fig. 8a–e**), as was the case for 5-FU–treated mice. RT-qPCR analysis revealed that several fetal intestine– and adult stomach–specific markers were markedly up-regulated in whole-crypt cells from irradiated mice compared with those from their normal counterparts (**new Supplementary Fig. 8f**). These results suggest that spatiotemporal reprogramming, including metaplasia-like changes, is a universal event common to intestinal injury contexts rather than being specific to 5-FU treatment. We have now described these results in the revised manuscript (page 13, lines 426–429).

Higa et al., Supplementary Figure 8

3. What happens if p57⁺ cells are ablated in different conditions, especially in conditions of damage? Seems like this question is readily answerable with the p57-DTR allele reported in this paper.

[Response] We generated p57-DTR-Venus mice in order to perform ablation of p57⁺ cells, and we confirmed that DT administration in these mice resulted in the efficient and specific elimination of p57-Venus⁺ cells (**Fig. R2a, for reviewing purposes only**). Unfortunately, however, we found that all of these mice die within 48 h after DT treatment, regardless of the absence or presence of intestinal injury, precluding long-term investigation of the effects of p57⁺ cell ablation. Of note, this mortality does not appear to be attributable to intestinal dysfunction, given the lack of histological abnormalities of the intestinal epithelium and weight loss, at least under the normal condition (data not shown). Given the high expression level of p57 in adrenal glands (Nagahama *et al.*, *Anat. Embryol.* 203: 77–87, 2001), we speculate that acute adrenal failure might be responsible for the mortality.

As a short-term analysis, we injected p57-DTR-Venus mice with DT at 3 days after 5-FU treatment and assessed crypt regeneration 24 h later (**Fig. R2b–e, for reviewing purposes only**) with the use of Ki67 immunostaining, as previously described (Metcalf *et al.*, *Cell Stem Cell* 14: 149–59, 2014). We found that the postinjury regenerative response apparent in control mice was markedly impaired in mice deficient in p57⁺ cells, as indicated by a significant decrease in the number of regenerating crypts. Although these results are suggestive of an indispensable role for crypt p57⁺ cells in postinjury epithelial regeneration, we cannot exclude the possibility that this phenotype was due to intestine- or epithelium-extrinsic effects, given that this experimental system results in the ablation of p57⁺ cells in all tissues and the mice were morbid. Taking these points into consideration, we decided not to include these data in the revised manuscript.

Higa et al. Figure R2

Figure R2 | **a** FACS plots for isolation of Venus⁺ cells from intestinal crypts either of p57-DTR-Venus mice injected with PBS or DT (50 μg/kg) 24 h before analysis or of WT mice as a negative control. **b** Dosing regimen for p57-DTR-Venus mice subjected to ablation of p57⁺ cells after the induction of crypt injury by injection of 5-FU (150 mg/kg). **c, d** Representative immunofluorescence staining of Ki67 in the intestinal epithelium at 4 days after injury induction without (**c**) or with (**d**) subsequent ablation of p57⁺ cells in p57-DTR-Venus mice. The active regenerative response apparent in **c** is impaired in **d**. Scale bars, 50 μm. **e** Box plots showing the number of viable crypts per field (1-mm length of the intestine) at 4 days after the induction of injury with or without subsequent p57⁺ cell ablation in p57-DTR-Venus mice as in **c** and **d**. Regenerating crypts were defined as cryptlike structures containing ≥5 adjacent Ki67⁺ cells, as previously described (Metcalf *et al.*, 2014). About 20 fields of view were counted per mouse (*n* = 3 mice). The box plots represent the maximum, upper quartile, lower quartile, and minimum values. Horizontal bars correspond to the median. *****P* < 0.001 (Student's *t* test).

To exclude the possible involvement of intestine- or epithelium-extrinsic effects, we next investigated the effect of p57⁺ cell ablation with the use of p57-Venus intestinal organoids *in vitro*. We found that combined treatment with 5-FU and DT resulted in marked impairment of organoid formation compared with 5-FU or DT treatment alone (**new Fig. 3l–n**). These results thus suggested that p57⁺ cells make a limited contribution to homeostatic epithelial

maintenance but are required for postinjury regeneration. We have now addressed these issues in the revised manuscript (pages 8–9, lines 244–265).

Higa et al., Figure 3

4. It would be appropriate to include appropriate controls for LGR5+ cell organoid formation (such as with ENRWC) – Figure 3 m,n.

[Response] As suggested by the reviewer, we quantified organoid formation efficiency for Lgr5^{high} CBCs under the ENRWC condition (**new Supplementary Fig. 3d**). The efficiency of CBC-derived organoid formation was significantly increased in ENRWC medium compared with ENR medium, likely as a result of augmentation of stemness or cell cycle activation by Wnt signaling rather than of induction of dedifferentiation. Of note, the data showing the Wnt-induced dedifferentiation of p57⁺ cells in **Figure 3l–n** of the original manuscript were moved to **new Supplementary Figure 3b–d** in the revised manuscript.

Higa et al., Supplementary Figure 3

5. There is a concern about the scRNA-seq data generated from this study. The data clustered by condition (Fig. 4) with no intermixing. It is unclear if the differences are generated by technical effects or biological effects. While the author stated it is $n=2$ per condition, it is unclear whether it is $n=2$ pooled together and ran as $n=1$ sample, or if it is $n=2$ independent experiments (separate runs). IT is important for multiple biological replicates to be performed independently to make sure the findings are reproducible and not resulting from technical effects.

[Response] We originally used $n = 2$ mice per condition but pooled them to obtain $n = 1$ biological replicate. In the revised manuscript, we repeated the same experiment with new mice, and the resultant scRNA-seq data (replicate 2) were analyzed together with the original data (replicate 1) in an integrated manner (**new Fig. 4**). We confirmed that spatiotemporal reprogramming and concomitant acquisition of stem cell-like transcriptomes were apparent in both of these replicates (**new Supplementary Fig. 4b**), thereby ensuring the reproducibility of our findings for the scRNA-seq analysis. We have now clarified the methodology and results for our new scRNA-seq analysis in both the Results and Methods sections of the revised manuscript (pages 9–11, lines 267–354; pages 21–23, lines 707–776).

With regard to the lack of intermixing of clusters across conditions in our original data, this was likely attributable to batch effects among scRNA-seq samples. In the revised manuscript, we analyzed the two replicates with the use of Harmony software to minimize unwanted batch effects. The intermixing of cells derived from different replicates or different conditions is now apparent in some clusters on the resultant t -SNE plot (**new Fig. 4b and Supplementary Fig. 4b**), suggesting that the cells are primarily clustered on the basis of biological differences. Without Harmony software, the cells were clearly clustered in accordance with sequencing batches (**new Supplementary Fig. 4c**). We have now addressed these points in the revised manuscript (page 9, lines 277–280).

Higa et al. Figure 4

Higa et al., Supplementary Figure 4

6. For RACEID2, please overlay on the same coordinates in figure 4. Unclear which cells belong to which condition. It's unclear whether they are different subsets of cells, but we could infer that they are different cell subsets since they seem to use two different filtering methods, one for *t*-SNE visualization and one for StemID. These filtering methods should be made consistent, if the comparison between the two *t*SNEs is going to be made. For example, there was a mitochondrial filter for Seurat-based *t*SNE but not for the StemID *t*SNE.

[Response] The StemID algorithm in principle calculates the score for each cluster identified by RaceID, not for each cell, and it is therefore impractical to overlay the results obtained by StemID analysis on the Seurat *t*-SNE plot. Instead of StemID, we decided to apply CytoTRACE software (ref. 39), which infers the degree of stemness for each single cell, so that we were then able to overlay the scores on the Seurat/Harmony *t*-SNE map (**new Fig. 5a**). As was the case for the StemID-based analysis in our original manuscript, postinjury p57⁺ cells showed a marked increase in the stemness score compared with their normal counterparts in the new analysis (**new Fig. 5b**). We have now addressed these issues in the revised manuscript (pages 11–12, lines 356–373).

Higa et al., Figure 5

7. The authors imply of Crohn's-like mechanism in metaplasia, which is shown to be epithelial cell-intrinsic with some overlaid signatures and histology. However, damage and inflammation are usually more complex than cell-intrinsic patterns and there should be at least some basic immune cell analyses to accompany the figures used in Figure 6.

[Response] Given that the activation of IFN- γ signaling was shown to contribute to the pathogenesis of intestinal inflammation (Langer *et al.*, *J. Clin. Invest.* 129: 4691–707, 2019) or infection (ref. 18), we performed an analysis of immune cells from this viewpoint. We found that the number of intestinal tissue-infiltrating neutrophils, which represent known targets of IFN- γ signaling (Amulic *et al.*, *Annu. Rev. Immunol.* 30, 459–89, 2012; ref. 18), was markedly increased in mice after 5-FU treatment compared with that in nontreated mice (**new Supplementary Fig. 7a–c**). In addition, the abundance of IFN- γ mRNA was significantly up-regulated in several types of lymphocyte including natural killer (NK) cells, T cells, and innate lymphoid cells (ILCs) 1/2 in mice treated with 5-FU (**new Supplementary Fig. 7d, e**). These results are consistent with those of previous studies implicating IFN- γ signaling in fetal-like reversion of the intestinal epithelium (ref. 18), and they indicate that activation of this pathway is likely also to underlie the metaplasia-like changes of the intestinal epithelium apparent after injury. We have now addressed these points in the revised manuscript (page 13, lines 412–417).

Higa et al., Supplementary Figure 7

8. *CDX2* and *CDX1* have some shared interactions, with respect to cellular plasticity. *CDX2* should also be highlighted or mentioned, as it has some implications in neoplasias.

[Response] Although we appreciate the reviewer's comment, *Cdx2* mRNA was virtually undetectable in our scRNA-seq data (**Fig. R3a, for reviewing purposes only**), making it difficult for us to discuss the role of this gene in cellular plasticity. In addition, we found that the down-regulation of *Cdx1* expression in postinjury p57⁺ cells, which had been mentioned in our original manuscript, was not obvious in our newly added scRNA-seq data (**Fig. R3b, for reviewing purposes only**). Although the DEG analysis for clusters 4/10/17/18 (consisting mostly of postinjury p57-Venus⁺ cells) versus all other clusters shown in **new Figure 4b** revealed a significant decrease in the abundance of *Cdx1* mRNA in the former

population (average $\log_2[\text{fold change}] = -0.61$, adjusted P value = 3.11×10^{-6}), we decided not to mention such a small change in the revised manuscript. With regard to transcription factors implicated in spatiotemporal reprogramming, we mention only *Nkx6-3* (**new Fig. 4g**), an inducer of metaplastic changes, and *Ascl2* (**new Fig. 4h**), which is encoded by a Wnt-regulated gene and confers stemness on intestinal cells. We hope the reviewer agrees with our decision not to describe the expression of *Cdx1/2*.

Higa et al. Figure R3

Figure R3 | **a, b** Heat maps showing the abundance of *Cdx2* (**a**) or *Cdx1* (**b**) mRNAs overlaid on the *t*-SNE projection.

Higa et al., Figure 4

Higa et al., Figure 4

Analysis comments:

9-1. Why, in 5c do there seem to be cell types that span multiple, seemingly disparate unsupervised clusters? This is the case for some enterocytes, paneth, CBC. Is there some signature difference between the different parts of these split-up clusters or potentially an artifact of Harmony batch correction?

[Response] In Figure 5c of our original manuscript, we performed an integrative analysis of our scRNA-seq data together with the data from Yan *et al.* (ref. 14) for the comparison of postinjury p57-Venus⁺ cells with normal crypt cells, given that our original scRNA-seq data contained a relatively small number of normal p57-Venus⁻ cells. In the revised manuscript, however, we omitted this analysis and instead provide scRNA-seq data for a larger number of normal p57-Venus⁻ cells (**new Fig. 4b and Supplementary Fig. 4b**). As mentioned above, we applied Harmony software for the scRNA-seq analysis in the revised manuscript to correct for batch effects.

In our new scRNA-seq analysis, we still found some coexistence of different cell types in individual clusters (**new Fig. 4b**). In particular, cluster 3 of our new scRNA-seq data contains both Paneth cells and secretory cells (probably consisting of common secretory progenitors and goblet lineage-committed cells), as judged from the expression patterns of cell type-specific markers (**new Supplementary Fig. 4e**). We consider that such clustering patterns result from both technical (algorithmic features of Harmony software) and biological (transcriptomic ambiguity of the cells) factors. The developers of Harmony software showed that it repeats soft clustering (explained as a quite fuzzy clustering, which assigns cells potentially to multiple clusters) until it reaches convergence, in order to obtain batch-corrected *t*-SNE plots (**Fig. R4, for reviewing purposes only**) (ref. 31). Whereas this algorithm is thought to be useful for classification of clearly distinct cell types with batch effects being corrected, we assume that such fuzzy clustering is likely to leave cells that possess “ambiguous” transcriptomes not clearly separated, compared with simple clustering tools such as Seurat. For example, cells halfway on their differentiation path are likely to possess bivalent transcriptomes reflecting two different cell states. Specifically, as mentioned above, cluster 3 of our new scRNA-seq data contains cells expressing both Paneth and secretory signatures at a high level (**new Supplementary Fig. 4e**), indicating that these cells are in an intermediate state between the goblet and Paneth lineage, or en route to differentiation from secretory precursors to Paneth cells. This could also be the case for CBCs and enterocytes. Overall, we conclude that the clustering patterns likely result from both Harmony-specific algorithmic features and biological ambiguity of cells with bivalent transcriptomes. Considering that each algorithm for scRNA-seq analysis has its own advantages and disadvantages, we decided largely to exploit the Harmony-computed *t*-SNE plots in our study, allowing us to discuss our findings with regard to cell type-based rather than batch effect-based clusters. We have now clarified these points in the revised manuscript (page 10, lines 286–288).

9-2. The authors should include some uncorrected scRNA-seq figures to compare.

[Response] In response to the reviewer’s comment, we now provide an uncorrected plot (**new Supplementary Fig. 4c**).

Higa et al., Figure 4

Higa et al., Supplementary Figure 4

Higa et al., Supplementary Figure 4

Higa et al. Figure R4

(cited from Figure 1 of Korsunsky et al., *Nat. Methods* 16: 1289–96, 2019)

Figure R4 | Overview of Harmony algorithm as shown in reference 31.

10. Some of the gene signatures examined have overlapping genes, so please consider the effect that these overlaps will have. The authors should ensure that these overlaps do not lead to misleading statistical test results.

[Response] As the reviewer points out, the fetal intestine signature and adult stomach signature contain three overlapping genes (*Gkn1*, *Psca*, and *Vsig1*). We found that the heat maps for each of these two signatures with or without the overlapping genes did not obviously differ. In the revised manuscript, we therefore calculated the expression level of these signatures with the overlapping genes removed (**new Fig. 4e, f**). We clarified the removal of overlapping genes between these two signatures in the annotation of **Supplementary Table 1**.

Higa et al., Figure 4

11. Figure 5a and b, entropy seems to be approximately between 1 and .5, why were these values not value scaled (i.e. start from 0), what is the nature of its distribution, and are these distributions significantly different by cluster or p57-Venus condition? I didn't see any statistical tests for these, just qualitative bar plots. Something like a violin plot per cluster or per sample type would be more interpretable.

[Response] As mentioned above, we used CytoTRACE software to calculate the stemness score for every single cell and omitted the data from StemID in the revised manuscript, rendering the reviewer's comment no longer relevant. This approach allowed us to clearly show cell-to-cell variation in the score on the *t*-SNE plots (**new Fig. 5a**) and to perform statistical analysis (**new Fig. 5b**). Postinjury p57-Venus⁺ cells had a significantly higher CytoTRACE score compared with normal p57-Venus⁺ cells or postinjury p57-Venus⁻ cells (**new Fig. 5b**), suggesting that they acquire stemness after injury and serve as a predominant stem cell population during regeneration. We have now clarified this point in the revised manuscript (pages 11–12, lines 356–373).

12. Figure 5a, the linkage edges are very difficult to interpret, why are some linked to ones across the tSNE? Is this due to some nonlinear reduction effect from tSNE and does this unusual linkage patterning maintain in PCA or UMAP space?

[Response] As mentioned above, our revised manuscript does not contain StemID lineage tree plots like Figure 5a of the original version. This comment is therefore no longer relevant.

13. Should show more than just Ki67 to the scRNA-seq analysis, consider published lists of different phase signatures, Seurat has a built-in cell cycle scoring that would strengthen the cell cycle claims for scRNA-seq figures.

[Response] As suggested by the reviewer, in addition to the original Ki67 expression plot (Supplementary Fig. 1c), we applied cell cycle scoring analysis to the scRNA-seq data in Figure 2 (new Supplementary Fig. 1d). We found that $Lgr5^{\text{high}}$ cells were divided into the G₁/S and G₂/M clusters, with Ki67 expression being higher in the latter population. We believe that these results reinforce the rationale for distinguishing cycling and noncycling CBC clusters. We have addressed this point in the revised manuscript (pages 5–6, lines 140–143).

Minor

- For Figure 1m, this could be better complemented by a cell cycle analysis instead of just ki67 staining.

[Response] As suggested by the reviewer, we reanalyzed our FACS data to evaluate the cell cycle status of Bmi1^{high} cells with the use of FlowJo v10.7.1 software. Although we found that the frequency of S/G₂/M-phase cells was higher for p57 CKO Bmi1^{high} cells compared with their WT counterparts in each experiment, this difference did not reach statistical significance (**Fig. R5, for reviewing purposes only**). We believe that this is likely attributable to the variability in PI staining among individual experiments, making it difficult to detect small changes in cell cycle status. As described in both the original and revised (page 5, lines 120–124) manuscripts, we instead performed EdU uptake analysis to verify cell cycle status in vivo. This assay revealed that the proportion of Bmi1^{high} cells in S phase was greater for the p57 CKO mouse intestine than for the WT control (**Fig. 1l, m**), suggesting that p57 gene deletion results in cell cycle activation in Bmi1^{high} cells.

Higa et al. Figure R5

Figure R5 | **a** Representative histogram plots of FACS analysis for PI staining of control or p57 CKO Bmi1-GFP^{high} crypt cells. Gating and the frequency of S/G₂/M-phase cells are shown. **b** Frequency of Bmi1-GFP^{high} crypt cells from control or p57 CKO mice in S/G₂/M phases of the cell cycle as quantified from experiments as in **a**. Data are means ± SEM ($n = 4$ mice). NS, not significant (Student's t test).

Higa et al., Figure 1

- Need to make Fig 2bd and 4 t-SNEs consistent in terms of dot scaling size.

[Response] As suggested by the reviewer, we have scaled the dots of scRNA-seq t -SNE plots to an equal size in the revised manuscript (**new Figs. 2 and 4**).

- Figure 5, could use a heatmap of scaled entropy scores per single cell. Difficult to interpret the 'lineage tree'.

[Response] As suggested by the reviewer, we calculated the 0-to-1 scaled CytoTRACE score for every single cell and overlaid it on the Harmony/Seurat *t*-SNE map in the revised manuscript (**new Fig. 5a, b**). We have now clarified this point in the revised manuscript (pages 11–12, lines 356–373).

- The authors should comment on the methodologies used to quantify the crypt numbers, like in Figure 6. Were they randomly selected fields of view or evaluated blind? Is 1+ cell enough to score etc.?

[Response] We apologize for not clarifying the detailed method for quantification of crypt number. We randomly selected the microscopic fields and counted the number of marker (*Tff2*, *Pgc*, or *Gif*)–positive crypts. Crypts containing at least three marker-positive cells were considered positive. The analysis was not blinded. We have now clarified these points in the legend of Figure 6 in the revised manuscript (page 34, lines 1095–1097).

- Fig 6a is not referenced anywhere, please ensure that all figures and subfigures are properly referenced.

[Response] We apologize for the error. We have now made sure that all figures and their panels are cited in the revised manuscript.

Response to Reviewer #3

We thank the reviewer for the careful assessment of our manuscript and for the statement that “Overall, this is an interesting study, which uses a number of elegant models to make some interesting observations on the cell type changes that occur during regeneration” and that “the precision with which the authors map the process certainly adds to our understanding of it and is a useful addition to the field.” We also thank the reviewer for several suggestions that we feel have helped us to improve our manuscript.

Our specific responses to the points raised are as follows:

1. The key take home message of the paper is that p57⁺ cells reprogramme and repopulate the intestine following injury. However, it is not clear to me how important p57⁺ cells are for regeneration as the authors do not directly test the consequences of depleting them in their models. This is important as in Figure 6 the authors suggest additional cell types also gain regenerative marker expression upon injury suggesting that many/all cell types in the intestine may have regenerative potential. It is possible that p57⁺ cells are simply a rare population of cells with no special properties different from other differentiated cell populations. The authors should investigate the functional consequences of depleting p57⁺ cells on the ability of the intestine to regenerate following injury. For example, they could utilise their p57-DTR model to deplete this cell population (alongside 5FU treatment or crossed to the Lgr5-DTR model).

[Response] As pointed out by the reviewer, ablation of p57⁺ cells is necessary in order to examine whether the regeneration and reprogramming of intestinal cells after injury are dependent on p57⁺ cells. We indeed generated p57-DTR-Venus mice in order to perform such ablation of p57⁺ cells, and we confirmed that DT administration resulted in the efficient and specific loss of p57-Venus⁺ cells in these mice (**Fig. R2a, for reviewing purposes only**). Unfortunately, however, we found that all such mice died within 48 h after DT treatment, regardless of the absence or presence of intestinal injury, thus precluding long-term investigation of the effects of p57⁺ cell ablation. Of note, this mortality did not appear to be attributable to intestinal dysfunction, given the lack of histological abnormalities of the intestinal epithelium and weight loss, at least under the normal condition (data not shown). Instead, we speculate that acute adrenal failure might be responsible for the mortality, given the high expression of p57 in the adrenal gland (Nagahama *et al.*, *Anat. Embryol.* 203: 77–87, 2001).

As a short-term analysis, we injected p57-DTR-Venus mice with DT 3 days after 5-FU treatment and assessed crypt regeneration 24 h later (**Fig. R2b–e, for reviewing purposes only**) with the use of Ki67 immunostaining, as previously described (Metcalf *et al.*, *Cell Stem Cell* 14: 149–59, 2014). We found that the postinjury regenerative response apparent in control mice was markedly impaired in mice deficient in p57⁺ cells, as indicated by a significant decrease in the number of regenerating crypts. Although these results are suggestive of an indispensable role for crypt p57⁺ cells in postinjury epithelial regeneration, we cannot exclude the possibility that this phenotype was the consequence of intestine- or epithelium-extrinsic effects, given that this experimental system results in the ablation of p57⁺ cells in all tissues and the mice were morbid. Taking these points into consideration, we decided not to include these data in the revised manuscript.

Higa et al. Figure R2

Figure R2 | **a** FACS plots for isolation of Venus⁺ cells from intestinal crypts either of p57-DTR-Venus mice injected with PBS or DT (50 µg/kg) 24 h before analysis or of WT mice as a negative control. **b** Dosing regimen for p57-DTR-Venus mice subjected to ablation of p57⁺ cells after the induction of crypt injury by injection of 5-FU (150 mg/kg). **c**, **d** Representative immunofluorescence staining of Ki67 in the intestinal epithelium at 4 days after injury induction without (**c**) or with (**d**) subsequent ablation of p57⁺ cells in p57-DTR-Venus mice. The active regenerative response apparent in **c** is impaired in **d**. Scale bars, 50 µm. **e** Box plots showing the number of viable crypts per field (1-mm length of the intestine) at 4 days after the induction of injury with or without subsequent p57⁺ cell ablation in p57-DTR-Venus mice as in **c** and **d**. Regenerating crypts were defined as cryptlike structures containing ≥5 adjacent Ki67⁺ cells, as previously described (Metcalfe *et al.*, 2014). About 20 fields of view were counted per mouse ($n = 3$ mice). The box plots represent the maximum, upper quartile, lower quartile, and minimum values. Horizontal bars correspond to the median. **** $P < 0.001$ (Student's *t* test).

To exclude the possible involvement of intestine- or epithelium-extrinsic effects, we next investigated the consequences of p57⁺ cell ablation with the use of p57-Venus intestinal organoids *in vitro*. We found that combined treatment with 5-FU and DT markedly impaired organoid formation compared with 5-FU or DT treatment alone (**new Fig. 3l–n**). These results thus suggested that p57⁺ cells make a limited contribution to homeostatic epithelial maintenance but are required for postinjury regeneration. We have now addressed these issues in the revised manuscript (pages 8–9, lines 244–265).

Higa et al., Figure 3

2. The single cell data (Fig 4c, d + e) suggests the majority of regenerative cells are derived from p57⁺ Venus⁺ cells. Why then is the regenerative potential of p57⁺ cells so low (Figure 3J - ~15 events per SI)?

[Response] We apologize for our data representation that might have been misleading. In our original data, we collected and subjected almost equal numbers of p57-Venus⁺ and p57-Venus⁻ cells to scRNA-seq analysis, given that we initially intended to investigate the postinjury transcriptomic changes of p57⁺ cells including their heterogeneity. The resultant plot therefore did not reflect the actual proportions of each cell type in intestinal crypts in vivo. Given that similar approaches have been widely adopted in previous scRNA-seq studies of tissue stem cells including ISCs, especially in order to investigate rare cell populations of interest (ref. 14; Pellin *et al.*, *Nat. Commun.* 10: 2395, 2019; Dong *et al.*, *Nat. Cell Biol.* 22: 630–39, 2020), we believe that our initial experimental design was not inappropriate. Nevertheless, we considered that such data representation might be likely to result in the misunderstanding that p57⁺ cells constitute the major cell population of intestinal crypts, as the reviewer points out. We therefore newly obtained scRNA-seq data from additional mouse samples (replicate 2), so that the number of p57⁻ cells was much larger than that of p57⁺ cells. In the revised manuscript, we analyzed these data together with our original data (replicate 1) in an integrated manner (**new Fig. 4 and Supplementary Fig. 4b**), and we confirmed that the key findings of this study (spatiotemporal reprogramming and postinjury stemness acquisition) are apparent in both of the replicates. We believe that this approach avoids extreme enrichment of p57⁺ cells as well as ensures the reproducibility of our findings. Although the proportion of p57⁺ cells on the *t*-SNE plot is still higher than that in actual intestinal crypts, we consider it impractical to perform the scRNA-seq analysis without enrichment of p57⁺ cells. We have now clarified the methodology and results of our new scRNA-seq analysis in both the Results and Methods sections of the revised manuscript (pages 9–11, lines 267–354; pages 21–22, lines 707–721).

Higa et al. Figure 4

Higa et al., Supplementary Figure 4

3. Fig 5 is an interesting analysis but would benefit from functional data. The authors could sort Venus⁺ cells from 5FU treated mice and determine whether they can form organoids *in vitro*.

[Response] As suggested by the reviewer, we subjected FACS-sorted intestinal p57-Venus⁺ cells from control and 5-FU-treated mice to organoid culture *in vitro*. We found that the postinjury p57-Venus⁺ cells form intestinal organoids, whereas the control p57-Venus⁺ cells almost completely lacked this ability (**new Fig. 5c, d**). These results suggest that p57-Venus⁺ cells indeed acquire stem cell potential during postinjury regeneration, consistent with the stemness inference analysis of our scRNA-seq data. We have now addressed this issue in the revised manuscript (page 12, lines 374–379).

Higa et al., Figure 5

4. Figure 1j-n characterises the effects of p57 ko on the dynamics of Bmi1⁺ cells suggesting it maintains their quiescence. Again, functional characterisation of these cells would be helpful – do they gain stem cell properties (organoid forming capacity) upon escape from quiescence?

[Response] In response to the reviewer's comment, we performed intestine-specific p57 gene deletion. We sorted Bmi1^{high} cells from WT and p57 CKO mice and subjected them to organoid culture. However, we did not observe organoid formation by either WT or p57 CKO Bmi1^{high} cells in ENR medium (data not shown). In addition, as an *in vivo* experiment to examine the effect of p57 gene deletion on Bmi1^{high} cells, we injected a single dose of tamoxifen (50 mg/kg) in Bmi1-CreER^{T2}/CAG-LSL-tdTomato/p57^{+/+} or Bmi1-

CreER^{T2}/CAG-LSL-tdTomato /p57^{+F} mice and traced Bmi1-tdTomato cells for 14 days. There was no significant difference in the frequency of Bmi1^{high} cell-derived lineage tracing events between these two mouse lines (**Fig. R6a, b, for reviewing purposes only**). Furthermore, in order to examine whether p57 gene deletion affects postinjury regeneration of the intestinal epithelium, we treated Villin-CreER^{T2}/p57^{+/+} (control) and Villin-CreER^{T2}/p57^{+F} (p57 CKO) mice with tamoxifen, injected them with 5-FU (150 mg/kg), and harvested the small intestine 3 days later. However, we did not detect any difference in Ki67 immunostaining (**Fig. R6c, for reviewing purposes only**), an indicator of postinjury regenerative activity in the intestinal epithelium (Metcalf *et al.*, *Cell Stem Cell* 14: 149–59, 2014), between the control and p57 CKO mouse lines. In addition, we did not observe any other intestinal phenotype such as weight loss or early mortality in these p57 CKO mice injected with 5-FU (data not shown).

These various data showed that p57 gene deletion did not affect the stem cell activity of Bmi1^{high} cells. We believe that this finding is not prejudicial with regard to the significance of our present study, given that we used p57 as a specific marker to trace the cells in which it is expressed, rather than focused on its molecular function in ISC regulation. Nevertheless, we think that the data showing that p57 maintains quiescence in Bmi1^{high} cells are worth including in our present manuscript, given that they provide a functional rationale for the use of p57 as a marker for quiescent cells around the crypt +4 position.

Higa et al. Figure R6

Figure R6 | **a** Representative immunofluorescence images of tdTomato in the small intestine of Bmi1-CreER^{T2}/CAG-LSL-tdTomato/p57^{+/+} (control) or Bmi1-CreER^{T2}/CAG-LSL-tdTomato/p57^{+F} (p57 CKO) mice injected with tamoxifen (50 mg/kg) and traced for 14 days. **b** Quantification of the number of Bmi1^{high} cell-derived lineage tracing events per 100 crypts determined from analysis as in **a**. Data are means + SEM ($n = 3$ mice). NS, not significant (Student's *t* test). **c** Representative immunofluorescence staining of Ki67 in the small intestine of Villin-CreER^{T2}/p57^{+/+} (control) or Villin-CreER^{T2}/p57^{+F} (p57 CKO) mice. The mice were treated with tamoxifen for 5 consecutive days to delete the p57 gene as in Figure 1j, injected with 5-FU (150 mg/kg), and analyzed 3 days after 5-FU injection.

REVIEWERS' COMMENTS

Reviewer #1 (Remarks to the Author):

The authors provided a very detailed response to all my previous comments. They added several new experiments including a new scRNA-Seq replicate. The revised manuscript has been substantially improved.

A comment on Supplement Fig. 2b, the authors stated that they tested Defa5 and Mmp7, instead of Lyz1, for Paneth cells, but did not clarify what was their technical issue or rationale of selecting these genes over the usage of Lyz1, Defa4, Defa6, which have been reported by other groups to have injury induced plasticity.

Reviewer #2 (Remarks to the Author):

The authors did a fine job addressing our concerns. The manuscript is generally much improved thru additional experiments, as well as corrected analyses. Some issues with batch correction still linger, but the authors did add more text explaining their limitations. We still have just a couple of minor comments for the authors.

1.

For the new cytotrace analysis, the authors showed very high values for p57 venus- normal cells. Why would that be the case since those cells would include both stem and differentiated cells (with low stem potential)?

2.

The authors should add to their discussion, and put into context their study with more current work describing metaplasia in human colonic tumor initiation (Chen et al., Cell, 2021).

Reviewer #3 (Remarks to the Author):

The authors have addressed my previous comments and I now recommend publication of this study.

Response to Reviewer #1

The authors provided a very detailed response to all my previous comments. They added several new experiments including a new scRNA-Seq replicate. The revised manuscript has been substantially improved.

We thank the reviewer for the careful appraisal of our revised manuscript and for the statements that “The authors provided a very detailed response to all my previous comments” and that “The revised manuscript has been substantially improved.” We also thank the reviewer for several suggestions that we feel have helped us to improve our manuscript.

Our specific responses to the points raised are as follows:

A comment on Supplement Fig. 2b, the authors stated that they tested Defa5 and Mmp7, instead of Lyz1, for Paneth cells, but did not clarify what was their technical issue or rationale of selecting these genes over the usage of Lyz1, Defa4, Defa6, which have been reported by other groups to have injury induced plasticity.

[Response] In the original manuscript, we have provided the data for Defa5 and Mmp7 expression in p57-Venus⁺ or p57-Venus⁻ cells simply because we had already tested them in our former experiments in a repeated fashion. We have routinely checked Defa5 and Mmp7 expression by RT-qPCR analysis according to a previous tracing study on intestinal LRCs (ref. 13), in which these genes were used for Paneth cell markers. As suggested by the reviewer, we have additionally examined the expression pattern of Lyz1, and obtained similar results to those of Defa5 and Mmp7. We have included the data to **new Supplementary Fig. 2b** in the revised manuscript.

Higa et al., Supplementary Figure 2

Response to Reviewer #2

The authors did a fine job addressing our concerns. The manuscript is generally much improved thru additional experiments, as well as corrected analyses. Some issues with batch correction still linger, but the authors did add more text explaining their limitations. We still have just a couple of minor comments for the authors.

We thank the reviewer for the careful appraisal of our revised manuscript and for the statements that “The authors did a fine job addressing our concerns” and that “The manuscript is generally much improved thru additional experiments, as well as corrected analyses.” We also thank the reviewer for several suggestions that we feel have helped us to improve our manuscript.

Our specific responses to the points raised are as follows:

1. For the new cytotracer analysis, the authors showed very high values for p57 venus-normal cells. Why would that be the case since those cells would include both stem and differentiated cells (with low stem potential)?

[Response] As the reviewer points out, the normal p57-Venus⁻ cells contain some differentiated cells with relatively high CytoTRACE scores, such as cluster 8 (enterocytes) or cluster 14 (secretory cells) (**Fig. 6a**). One possible explanation for this is that these clusters might contain relatively immature enterocyte or secretory progenitors, given that we mainly collected crypts and almost completely removed villi for the scRNA-seq sample preparation. In postinjury samples, however, these clusters are almost completely undetectable, indicating that they were lost upon injury as is the case in Lgr5^{high} CBCs, presumably due to their active cell cycle status. We speculate that, after the loss of such immature or actively cycling cells, more differentiated or slowly cycling cell types including p57⁺ cells serve as an important source for injury-induced ISCs through the reprogramming process. Although this is an important and interesting point, we hope the reviewer agrees with our decision not to discuss these points in detail in this manuscript, given that we are just able to make a speculation as above and its experimental validation is beyond the scope of this study.

Higa et al., Figure 6

2. The authors should add to their discussion, and put into context their study with more current work describing metaplasia in human colonic tumor initiation (Chen et al., Cell, 2021).

[Response] As suggested by the reviewer, we have cited the study describing metaplasia in human colonic tumor initiation (ref. 47) in the revised manuscript (page 16, lines 513-515).

Response to Reviewer #3

The authors have addressed my previous comments and I now recommend publication of this study.

We thank the reviewer for the careful appraisal of our revised manuscript and for the statements that “The authors have addressed my previous comments and I now recommend publication of this study”. This reviewer has no further critiques.